# A COORDINATE-FREE CONSTRUCTION OF SCALABLE NATURAL GRADIENT

## ABSTRACT

Most neural networks are trained using first-order optimization methods, which are sensitive to the parameterization of the model. Natural gradient descent is invariant to smooth reparameterizations because it is defined in a coordinate-free way, but tractable approximations are typically defined in terms of coordinate systems, and hence may lose the invariance properties. We analyze the invariance properties of the Kronecker-Factored Approximate Curvature (K-FAC) algorithm by constructing the algorithm in a coordinate-free way. We explicitly construct a Riemannian metric under which the natural gradient matches the K-FAC update; invariance to affine transformations of the activations follows immediately. We extend our framework to analyze the invariance properties of K-FAC appied to convolutional networks and recurrent neural networks, as well as metrics other than the usual Fisher metric.

## 1 INTRODUCTION

Most neural networks are trained using stochastic gradient descent (SGD) (Bottou & Bousquet, 2007), or variants thereof which adapt step sizes for individual dimensions (Duchi et al., 2011; Kingma & Ba, 2015). One well-known deficiency of SGD is that the updates are sensitive to the parameterization of the network. There are numerous tricks for reparameterizing network architectures so that they represent the same sets of functions, but in a friendlier coordinate system. Examples include replacing logistic activation functions with $\tanh$ (Glorot & Bengio, 2010), whitening the inputs or activations (Desjardins et al., 2015; Krizhevsky, 2009), or centering the activations to have zero mean and/or unit variance (Montavon & Müller, 2012; Cho et al., 2011; Raiko et al., 2012; Ioffe & Szegedy, 2015). Such tricks can lead to large improvements in the speed of optimization.

Ideally, one would like to use an optimization algorithm which is invariant to such transformations of a neural network, in order to avoid the pathologies which the transformations are meant to remedy. Natural gradient descent (Amari, 1998) is a second-order optimization algorithm motivated by a key invariance property: to the first-order, its updates are invariant to smooth reparameterizations of a model. The natural gradient of a cost function can be seen as the gradient of the function on a Riemannian manifold (typically using the Fisher information metric (Amari & Nagaoka, 2000)), and the invariance properties of the algorithm follow directly from its definition in terms of differential geometric primitives.

There have been many attempts to apply natural gradient descent, or approximations thereof, to training neural networks (Amari et al., 2000; Roux et al., 2008; Martens, 2010; Grosse & Salakhutdinov, 2015; Martens & Grosse, 2015; Desjardins et al., 2015; Schulman et al., 2015). The challenge is that the exact natural gradient is impractical to compute for large neural nets, because it requires solving a linear system whose dimension is the number of parameters (which may be in the tens of millions for modern networks). Unfortunately, tractable approximations to the natural gradient are typically defined in terms of particular coordinate representations, and therefore may lose the invariance properties which motivated natural gradient in the first place.

Kronecker-Factored Approximate Curvature (K-FAC) (Martens & Grosse, 2015) is an approximate natural gradient optimizer where the Fisher information matrix $\mathbf{F}$ is approximated as a block diagonal matrix with one block per layer of the network, and each block factorizes as the Kronecker product of small matrices. Because of the Kronecker structure, the approximate natural gradient can be computed with low overhead relative to ordinary SGD; K-FAC demonstrated significant

speedups in training deep autoencoders (Martens & Grosse, 2015), classification convolutional networks (Grosse & Martens, 2016; Ba et al., 2017), recurrent networks (Martens et al., 2018) and deep reinforcement learning (Wu et al., 2017). The same Fisher matrix approximation has also led to significant improvements in modeling posterior uncertainty in Bayesian neural networks (Ritter et al., 2018b; Zhang et al., 2017) and avoiding catastrophic forgetting (Ritter et al., 2018a). Most recently, K-FAC was used to train a neural network to solve the many-electron Schrödinger equation (Pfau et al., 2019).

Although K-FAC does not satisfy the general invariance properties of natural gradient, it is still invariant to a broad and interesting class of reparameterizations: affine transformations of the activations in each layer (Martens & Grosse, 2015). This was verified through linear algebraic manipulation of the update rules, but unfortunately the proofs yielded little insight into the algorithm or advice about how it can be extended.

In this paper, we take a different approach: we formulate K-FAC directly in terms of coordinate-free mathematical objects, so that the invariance properties follow immediately from the construction. Specifically, we view a neural network as a series of affine maps alternating with fixed nonlinear activation functions. The activations and pre-activations for each layer are viewed as abstract affine spaces, and the weights and biases of the network correspond to affine maps. The ordinary Fisher metric is a metric on this space $\mathcal{W}$ of affine maps. Our contribution is a recipe to convert a metric on $\mathcal{W}$ (whose coordinate representation is extremely large) into an approximate metric on $\mathcal{W}$ (the "*K-FAC metric*"), whose coordinate representation matches the K-FAC approximation. Hence, rather than view K-FAC as an approximation to the natural gradient under the Fisher metric, we view it as the exact natural gradient under the K-FAC metric. This entire construction is coordinate-free, so the invariance properties of K-FAC follow immediately.

We can contrast K-FAC's invariance properties with those of exact natural gradient descent. Since the exact natural gradient is derived in terms of a metric on a smooth manifold, the update is invariant to arbitrary smooth reparameterizations, but only up to the first-order. By contrast, we show global invariance to a more restricted class of reparameterizations. Our analysis imposes additional structure on the weight manifold $\mathcal{W}$: the parameters are assumed to define affine maps between affine spaces. Choosing affine bases for the activations and pre-activations yields a natural affine basis for $\mathcal{W}$. Therefore, the set of allowable reparameterizations for neural networks consists of affine change-of-basis transformations for the activations and pre-activations. This leaves out some unusual reparameterizations which exact natural gradient descent is invariant to, such as permuting the entries of the weight matrix. But it captures important classes of reparameterizations, such as whitening, normalization, and replacing the logistic activation function with $\tanh$. And in exchange for imposing the affine structure, we obtain *global* invariance, not just first-order invariance.

Our framework easily enables some generalizations of the basic result. Our construction is not limited to Fisher metrics, but rather it applies to general *pullback metrics*, where one places a metric on the network's output space and pulls it back to $\mathcal{W}$. Furthermore, we extend the invariance results to convolutional networks and recurrent networks through a straighforward application of our K-FAC metric construction, highlighting the versatility of adopting the coordinate-free approach.

**Notations.** Since our coordinate-free construction of K-FAC requires mathematical machinery from both abstract affine/linear algebra and differential geometry, we provide an overview of necessary concepts in Appendix B. As we go back-and-forth between coordinate-independent and coordinate-dependent objects frequently in this paper, we provide a summary of all notations in Appendix A. The $[\![\cdot]\!]$ notation throughout stands for choosing coordinates for an abstract mathematical object.

## 2 BACKGROUND

### 2.1 NATURAL GRADIENT DESCENT

For simplicity, we define the natural gradient descent algorithm here in the context of multilayer perceptrons (MLPs), i.e., fully connected feed-forward networks. For an input-target pair $(\mathbf{x}, \mathbf{y})$, let $f(\mathbf{x}, \mathbf{w})$ denote the output and $\mathbf{w}$ symbolizes the parameter vector of the MLP. Given a training set

$\mathcal{S}$ of pairs $(\mathbf{x}_i, \mathbf{y}_i)$, we aim to minimize the empirical risk which is given by:

$$h(\mathbf{w}) = \frac{1}{|\mathcal{S}|} \sum_{(\mathbf{x}_i, \mathbf{y}_i) \in \mathcal{S}} \mathcal{L}(\mathbf{y}_i, f(\mathbf{x}_i, \mathbf{w})), \tag{2.1}$$

where $\mathcal{L}(\mathbf{y}_i, f(\mathbf{x}_i, \mathbf{w}))$ is the loss function measuring the disagreement between $\mathbf{y}_i$ and $f(\mathbf{x}_i, \mathbf{w})$.

Suppose that $f(\mathbf{x}, \mathbf{w})$ determines parameters $\mathbf{z}$ of the model's predictive distribution $R_{\mathbf{y}|\mathbf{z}}$ over $\mathbf{y}$ and furthermore, we reparameterize this as $P_{\mathbf{y}|\mathbf{x}}(\mathbf{w}) = R_{\mathbf{y}|f(\mathbf{x},\mathbf{w})}$. Likewise, the density function of this distribution can be reparameterized as $p(\mathbf{y}|\mathbf{x}, \mathbf{w}) = r(\mathbf{y}|f(\mathbf{x}, \mathbf{w}))$. In addition, we take the loss function here to be the negative log-likelihood $\mathcal{L}(\mathbf{y}, \mathbf{z}) = -\log r(\mathbf{y}|\mathbf{z})$ and denote log-likelihood gradients $\nabla_{\mathbf{w}} \mathcal{L}(\mathbf{y}, f(\mathbf{x}, \mathbf{w}))$ by $\mathcal{D}\mathbf{w}$ ($\mathcal{D}$ notation throughout remainder of the paper refers to log-likelihood gradients). The Fisher information matrix $\mathbf{F}(\mathbf{w})$ is defined as

$$\mathbf{F}(\mathbf{w}) = \mathbb{E}_{\mathbf{x},\mathbf{y}}[(\mathcal{D}\mathbf{w})(\mathcal{D}\mathbf{w})^\top], \tag{2.2}$$

where the expectation is taken over $P_{\mathbf{y}|\mathbf{x}}(\mathbf{w})$ for $\mathbf{y}$ and over the data distribution for $\mathbf{x}$. Since $\mathbf{F}(\mathbf{w})$ is defined as the expectation of an outer product, $\mathbf{F}(\mathbf{w})$ is always guaranteed to be a positive-semidefinite (PSD) matrix.

The natural gradient of the objective function $h(\mathbf{w})$ in Eqn. 2.1 is $\mathbf{F}(\mathbf{w})^{-1} \nabla h(\mathbf{w})$. For a chosen learning rate $\epsilon > 0$, the natural gradient descent algorithm (Amari, 1998) minimizes $h(\mathbf{w})$ by using the natural gradient to update parameters of the network:

$$\mathbf{w}_{k+1} \leftarrow \mathbf{w}_k - \epsilon \mathbf{F}(\mathbf{w}_k)^{-1} \nabla h(\mathbf{w}_k). \tag{2.3}$$

**Invariance Properties of Natural Gradient.** Natural gradient descent possesses a key invariance property which does not hold for ordinary stochastic gradient descent (SGD): given two equivalent networks which are parameterized differently, after applying the natural gradient descent update to each, the resulting networks will be equivalent up to the first-order. The reason for this is that the natural gradient admits an intrinsic coordinate-free construction in terms of differential geometric primitives. We give a detailed explanation of this by considering an abstract mathematical setting; we defer to Appendix B for background on the mathematical terminology with which we use.

Let $\mathcal{M}$ be a Riemannian manifold with nondegenerate metric given by $g$ (in this paper, our definition of "metrics" allows the possibility of it being degenerate). For a smooth function $h : \mathcal{M} \to \mathbb{R}$ and a point $p \in \mathcal{M}$, the differential $dh(p)$ is an abstract covector on $\mathcal{M}$. We use the nondegenerate metric $g$ to convert the covector $dh(p)$ into a tangent vector. By definition, $g(p)$ is a nondegenerate bilinear form which yields a linear isomorphism between the tangent space and the cotangent space:

$$g(p) : T_p\mathcal{M} \xrightarrow{\cong} T_p^*\mathcal{M}.$$

This is commonly referred to as the musical isomorphism in mathematical literature. The inverse $g(p)^{-1}$ gives a linear map the other way around,

$$g(p)^{-1} : T_p^*\mathcal{M} \xrightarrow{\cong} T_p\mathcal{M}.$$

Applying this isomorphism to $dh(p)$ yields the tangent vector $g(p)^{-1} dh(p) \in T_p\mathcal{M}$. We call this tangent vector the natural gradient of $h$.

We apply this mathematical framework to the objects of our interest. First, let $\mathcal{W}$ be a smooth manifold which characterizes the weight space of network parameters intrinsically. Let $\omega$ and $(\xi, \upsilon)$ be intrinsic versions of the parameter vector $\mathbf{w}$ and the input-target pair $(\mathbf{x}, \mathbf{y})$ respectively. $\xi$ and $\upsilon$ belong to abstract input and output spaces which are also smooth manifolds. Now, $\mathcal{W}$ can be endowed with the Fisher metric $g_F$ which is defined as

$$g_F(\omega) = \mathbb{E}_{\xi,\upsilon}[d\mathcal{L}_\omega \otimes d\mathcal{L}_\omega], \tag{2.4}$$

where $\mathcal{L}_\omega = -\log p(\upsilon|\xi, \omega)$ is the abstract log-likelihood loss function. The expectation above is taken over the abstract predictive distribution $P_{\upsilon|\xi}(\omega)$ for $\upsilon$ and over the data distribution for $\xi$. Expressing this in a coordinate system, we obtain exactly the Fisher matrix as given in Eqn. 2.2.

As $\mathcal{W}$ is a Riemannian manifold with the Fisher metric $g_F$, the idealized gradient descent updates are given by Bonnabel et al. (2013)

$$\omega_{k+1} \leftarrow \text{Exp}_{\omega_k}(-\epsilon g_F(\omega_k)^{-1} dh(\omega_k)), \tag{2.5}$$

where $\mathrm{Exp}_{\omega_k} : T_{\omega_k}\mathcal{W} \to \mathcal{W}$ is the exponential map. This update rule is *exactly* invariant to all smooth reparameterizations of $\mathcal{W}$ since it is entirely coordinate-free. However, such an algorithm is infeasible in practice as computing the exponential map is typically an intractable problem. Instead, it is much easier to work with the following abstract natural gradient update rule which uses a first-order approximation of the exponential map

$$\omega_{k+1} \leftarrow -\epsilon g_F(\omega_k)^{-1} dh(\omega_k).$$

Since this is a first-order approximation of the update rule in Eqn. 2.5, invariance to smooth reparameterizations holds only up to first-order. Additional approximations to the exponential map are necessary to obtain higher-order invariances; we defer to Song & Ermon (2018) for a more detailed account of how this can be done.

## 2.2 KRONECKER-FACTORED APPROXIMATE CURVATURE

We consider a MLP with $L$ layers. At each layer $i \in \{1, \ldots, L\}$, the MLP computation is given as:

$$\mathbf{z}_i = \mathbf{W}_i \mathbf{a}_{i-1} + \mathbf{b}_i$$
$$\mathbf{a}_i = \phi_i(\mathbf{z}_i),$$

where $\mathbf{a}_{i-1}$ is an activation vector, $\mathbf{z}_i$ is a pre-activation vector, $\mathbf{W}_i$ is a weight matrix, $\mathbf{b}_i$ is a bias vector, and $\phi_i : \mathbb{R} \to \mathbb{R}$ is an activation function. For convenience, we introduce homogeneous coordinates $\bar{\mathbf{a}}_{i-1}^\top = [\mathbf{a}_{i-1}^\top \ 1]^\top$ and $\bar{\mathbf{W}}_i = [\mathbf{W}_i \ \mathbf{b}_i]$. The above computation can be rewritten as

$$\mathbf{z}_i = \bar{\mathbf{W}}_i \bar{\mathbf{a}}_{i-1}$$
$$\mathbf{a}_i = \phi_i(\mathbf{z}_i). \tag{2.6}$$

We concatenate all of the network parameters $\bar{\mathbf{W}}_i$ into a single vector $\mathbf{w}$,

$$\mathbf{w} = [\mathrm{vec}(\bar{\mathbf{W}}_1)^\top \ \mathrm{vec}(\bar{\mathbf{W}}_2)^\top \ \ldots \ \mathrm{vec}(\bar{\mathbf{W}}_L)^\top]^\top.$$

Here, $\mathrm{vec}$ denotes the vectorization operator which stacks the columns of a matrix together to form a vector. The Fisher matrix for the MLP is a $L \times L$ block matrix $\mathbf{F}(\mathbf{w})$ where each $(i, j)$-th block is

$$\mathbf{F}(\mathbf{w})_{i,j} = \mathbb{E}[\mathrm{vec}(\mathcal{D}\bar{\mathbf{W}}_i) \mathrm{vec}(\mathcal{D}\bar{\mathbf{W}}_j)^\top].$$

We now give an overview of the Kronecker-Factored Approximate Curvature (K-FAC) method (Martens & Grosse, 2015) of approximating the Fisher matrix. Consider the diagonal $(i, i)$ blocks of $\mathbf{F}(\mathbf{w})$. Using backpropagation, the log-likelihood gradient $\mathcal{D}\bar{\mathbf{W}}_i = \mathcal{D}\mathbf{z}_i \bar{\mathbf{a}}_{i-1}^\top$ and hence, we have $\mathrm{vec}(\mathcal{D}\mathbf{z}_i \bar{\mathbf{a}}_{i-1}^\top) = \bar{\mathbf{a}}_{i-1} \otimes \mathcal{D}\mathbf{z}_i$. Then, $\mathbf{F}(\mathbf{w})_{i,i}$ can be rewritten as:

$$\mathbf{F}(\mathbf{w})_{i,i} = \mathbb{E}[\mathrm{vec}(\mathcal{D}\bar{\mathbf{W}}_i) \mathrm{vec}(\mathcal{D}\bar{\mathbf{W}}_i)^\top] = \mathbb{E}[(\bar{\mathbf{a}}_{i-1} \otimes \mathcal{D}\mathbf{z}_i)(\bar{\mathbf{a}}_{i-1} \otimes \mathcal{D}\mathbf{z}_i)^\top] = \mathbb{E}[\bar{\mathbf{a}}_{i-1}\bar{\mathbf{a}}_{i-1}^\top \otimes \mathcal{D}\mathbf{z}_i \mathcal{D}\mathbf{z}_i^\top],$$

where $\otimes$ denotes the Kronecker product of matrices. If the activations and pre-activation derivatives are approximated as independent, this yields the following approximation $\hat{\mathbf{F}}(\mathbf{w})_{i,i}$ to $\mathbf{F}(\mathbf{w})_{i,i}$,

$$\hat{\mathbf{F}}(\mathbf{w})_{i,i} = \underbrace{\mathbb{E}[\bar{\mathbf{a}}_{i-1}\bar{\mathbf{a}}_{i-1}^\top]}_{:=\mathbf{A}_{i-1}} \otimes \underbrace{\mathbb{E}[\mathcal{D}\mathbf{z}_i \mathcal{D}\mathbf{z}_i^\top]}_{:=\mathbf{G}_i} \tag{2.7}$$

The K-FAC approximation matrix $\hat{\mathbf{F}}(\mathbf{w})$ to $\mathbf{F}(\mathbf{w})$ is defined as

$$\hat{\mathbf{F}}(\mathbf{w}) = \begin{bmatrix} \mathbf{A}_0 \otimes \mathbf{G}_1 & & & \mathbf{0} \\ & \mathbf{A}_1 \otimes \mathbf{G}_2 & & \\ & & \ddots & \\ \mathbf{0} & & & \mathbf{A}_{L-1} \otimes \mathbf{G}_L \end{bmatrix}. \tag{2.8}$$

The K-FAC update rules are analogous to natural gradient descent updates; we simply replace $\mathbf{F}(\mathbf{w}_k)^{-1}$ in Eqn. 2.3 by $\hat{\mathbf{F}}(\mathbf{w}_k)^{-1}$ above.

**Invariance Properties of K-FAC.** Since K-FAC uses the approximation $\hat{\mathbf{F}}(\mathbf{w})$ rather than the Fisher matrix $\mathbf{F}(\mathbf{w})$ itself, the invariance properties of natural gradient do not necessarily carry over to K-FAC. Instead, we consider the class of transformations given by the following transformed network

$$\mathbf{z}_i^\dagger = \bar{\mathbf{W}}_i^\dagger \bar{\mathbf{a}}_{i-1}^\dagger$$
$$\mathbf{a}_i^\dagger = \phi_i^\dagger(\mathbf{z}_i^\dagger) = \mathbf{\Omega}_i \phi_i(\mathbf{\Phi}_i \mathbf{z}_i + \boldsymbol{\tau}_i) + \boldsymbol{\gamma}_i, \tag{2.9}$$

where $\mathbf{\Omega}_i$, $\mathbf{\Phi}_i$ are invertible matrices and $\boldsymbol{\tau}_i$, $\boldsymbol{\gamma}_i$ are vectors. The transformed input is $\bar{\mathbf{a}}_0^\dagger = \bar{\mathbf{\Omega}}_0 \bar{\mathbf{a}}_0$ where

$$\bar{\mathbf{\Omega}}_0 = \left[ \begin{array}{cc} \mathbf{\Omega}_0 & \boldsymbol{\gamma}_0 \\ \mathbf{0} & 1 \end{array} \right],$$

and the transformed output is $\mathbf{a}_L^\dagger = f^\dagger(\mathbf{x}^\dagger, \mathbf{w}^\dagger)$ with $\mathbf{w}^\dagger$ defined as

$$\mathbf{w}^\dagger = [\text{vec}(\bar{\mathbf{W}}_1^\dagger)^\top \ \text{vec}(\bar{\mathbf{W}}_2^\dagger)^\top \ \ldots \ \text{vec}(\bar{\mathbf{W}}_L^\dagger)^\top]^\top.$$

The original and transformed network are equivalent in terms of the functions they compute. We observe that the transformations given in Eqn. 2.9 encompasses a wide range of transformations. These include common deep learning tricks such as centering the activations to have zero mean and/or unit variance and replacing logistic sigmoid activation functions with $\tanh$. While K-FAC may not be invariant under smooth parameterizations of the model as in the case of natural gradient, the following theorem shows that it is invariant to the class of transformations given in Eqn. 2.9.

**Theorem 2.1.** *Let $\mathcal{N}$ be the network with parameter vector $\mathbf{w}$ and activation functions $\{\phi_i\}_{i=1}^L$. Suppose that we have activation functions $\{\phi_i^\dagger\}_{i=1}^L$ as given in Eqn. 2.9. Then, there exists a parameter vector $\mathbf{w}^\dagger$ such that the transformed network $\mathcal{N}^\dagger$ with parameter vector $\mathbf{w}^\dagger$ and activation functions $\{\phi_i^\dagger\}_{i=1}^L$ computes the same function as $\mathcal{N}$. Furthermore, if the Fisher matrix of the MLP is assumed to be positive-definite, then the K-FAC updates are equivalent, in the sense that the resulting networks compute the same function.*

A proof of this result was given earlier in Martens & Grosse (2015); however, the proof was cumbersome and involved tedious manipulation of update rules. In contrast, we like to provide a far more mathematically mature and elegant way to obtain this result. That is, we construct an intrinsic K-FAC metric $g_{\text{KFAC}}$ whose coordinate representation exactly matches the K-FAC approximation given in Eqn. 2.8. A K-FAC update can then be viewed as a natural gradient update with respect to the metric $g_{\text{KFAC}}$. More importantly, the invariance properties of the K-FAC algorithm are immmediately established in the same way as it was for exact natural gradient.

## 3 COORDINATE-FREE K-FAC

We observe that MLPs consist of a sequence of affine transformations and activation functions in alternation. In order to capture this structure, we treat the spaces of activations and pre-activations as affine spaces. Note that this introduces more structure than was assumed when we discussed the exact natural gradient in Section 2.1; in that subsection, we treated the space of network parameters as a general smooth manifold. Here, the network weights and biases are assumed to define affine transformations. The set of allowable reparameterizations (and hence, the desired set of invariances) is correspondingly more limited (though still very broad). We now present the coordinate-free MLP formally. For $i \in \{1, \ldots, L\}$, we have

- Activations are taken to be elements $\alpha_{i-1}$ in an affine space $\mathcal{A}_{i-1}$.
- Pre-activations are taken to be elements $\zeta_i$ in an affine space $\mathcal{Z}_i$.
- Layerwise parameters are affine transformations $\omega_i$ between $\mathcal{A}_{i-1}$ and $\mathcal{Z}_i$. The collection of these transformations is an affine space in its own right, which we denote by $\mathcal{W}_i$ and refer to as the layerwise weight space.
- The weight space is given by the direct product $\mathcal{W} = \mathcal{W}_1 \times \cdots \times \mathcal{W}_L$. Elements in this space are written as $\omega = (\omega_1, \ldots, \omega_L) \in \mathcal{W}$.
- Input and outputs are denoted by $\xi$ and $f(\xi, \omega)$ respectively. The space of all inputs and outputs are affine spaces denoted by $\mathcal{X}(= \mathcal{A}_0)$ and $\mathcal{Y}(= \mathcal{A}_L)$ respectively.

Moreover, the layerwise computation is given by

$$\begin{aligned} \zeta_i &= \omega_i(\alpha_{i-1}) \\ \alpha_i &= \rho_i(\zeta_i), \end{aligned} \tag{3.1}$$

where $\rho_i : \mathbb{R} \to \mathbb{R}$ is a fixed nonlinear activation function which is assumed to be smooth throughout. In Appendix D, we show that the MLP with computation defined in Eqn. 2.6 and the transformed version in Eqn. 2.9 correspond to two different choices of parameterizations of the same underlying abstract MLP.

*Remark* 3.1. The smoothness condition on the activation functions $\rho_i$ is necessary as our later analysis is built upon the framework of differential geometry. Smoothness is required to define standard operations such as pushforwards and pullbacks. While the commonly-used ReLU activation function is not smooth by definition, it is locally smooth with probability 1 if the weights are sampled from a continuous distribution. Furthermore, the ReLU function is a limit of the softplus functions $\frac{1}{n}\log(1 + e^{nx})$, which are all individually smooth.

**Optimization.** The optimization problem in the abstract setting is analogous to the coordinate-dependent one. Let $(\xi, \upsilon)$ be an abstract input-target pair and $\mathcal{L}(\upsilon, f(\xi, \omega))$ be the loss function measuring the disagreement between outputs $f(\xi, \omega)$ of the abstract MLP and targets $\upsilon$. Given a training set $\mathcal{S}$ of abstract input-target pairs $(\xi_i, \upsilon_i)$, the objective function we wish to minimize here is

$$h(\omega) = \frac{1}{|\mathcal{S}|} \sum_{(\xi_i, \upsilon_i) \in \mathcal{S}} \mathcal{L}(\upsilon_i, f(\xi_i, \omega)).$$

The following theorem (see Appendix E.1 for the proof) shows that by imposing affine structure on the weight space, we can obtain *global* invariance instead of just first-order invariance.

**Theorem 3.2.** *Let $g$ be a nondegenerate metric on the weight space $\mathcal{W}$ of an abstract MLP. For a chosen learning rate $\epsilon > 0$, the following update rule*

$$\omega_{k+1} \leftarrow \omega_k - \epsilon g(\omega_k)^{-1} dh(\omega_k),$$

*is* exactly *invariant to all affine reparameterizations of the model.*

**Pullback of Output Metrics to Weight Space.** Consider a metric $g$ on the output space $\mathcal{Y}$ of the MLP. Let $\Psi_\xi : \mathcal{W} \to \mathcal{Y}$ be the smooth map which sends parameters $\omega$ to outputs $\alpha_L = f(\xi, \omega)$ given an input $\xi$. The pullback $\Psi_\xi^* g$ defines a metric on $\mathcal{W}$. The expected pullback metric over inputs, under a choice of coordinates around $\omega$ and $\alpha_L$, is given by

$$[\![\mathbb{E}_\xi[\Psi_\xi^* g(\omega)]]\!] = \mathbb{E}_{\mathbf{x}}[\mathbf{J}_{\Psi_\xi}^\top \mathbf{G} \mathbf{J}_{\Psi_\xi}], \tag{3.2}$$

where $\mathbf{G}$ is the representation of $g$ in these coordinates.

**Example 3.3** (Fisher metric). Suppose that the outputs $\alpha_L$ parameterize the model's predictive distribution $R_{\upsilon|\alpha_L}$. Let $r(\upsilon|\alpha_L)$ denote the density function of this distribution and furthermore, we take the loss function here to be the negative log-likelihood $\mathcal{L}_{\alpha_L} = -\log r(\upsilon|\alpha_L)$. The output Fisher metric $g_{F,\text{out}}$ on $\mathcal{Y}$ is defined as

$$g_{F,\text{out}}(\alpha_L) = \mathbb{E}_\upsilon[d\mathcal{L}_{\alpha_L} \otimes d\mathcal{L}_{\alpha_L}],$$

where the expectation is taken with respect to the predictive distribution $R_{\upsilon|\alpha_L}$. Computing the expectation of $\Psi_\xi^* g_{F,\text{out}}$ over the inputs $\xi$ gives

$$\begin{aligned}
\mathbb{E}_\xi[\Psi_\xi^* g_{F,\text{out}}(\omega)] &= \mathbb{E}_\xi[\Psi_\xi^* \mathbb{E}_\upsilon[d\mathcal{L}_{\alpha_L} \otimes d\mathcal{L}_{\alpha_L}]] \\
&= \mathbb{E}_{\xi,\upsilon}[\Psi_\xi^*(d\mathcal{L}_{\alpha_L} \otimes d\mathcal{L}_{\alpha_L})] \\
&= \mathbb{E}_{\xi,\upsilon}[\Psi_\xi^*(d\mathcal{L}_{\alpha_L}) \otimes \Psi_\xi^*(d\mathcal{L}_{\alpha_L})] \\
&= \mathbb{E}_{\xi,\upsilon}[d\mathcal{L}_\omega \otimes d\mathcal{L}_\omega].
\end{aligned}$$

This is exactly the Fisher metric defined earlier in Eqn. 2.4.

We note that this construction quite general and not limited to Fisher metrics; for example, if we use the Euclidean metric on $\mathcal{Y}$, then Eqn. 3.2 is the Gauss-Newton matrix (Martens, 2014). A practical use case for the Gauss-Newton metric is when the outputs of the network do not have a natural probabilistic interpretation, e.g., the value network in an actor-critic architecture for reinforcement learning (Wu et al., 2017). Another example is if we take the Riemannian metric induced by the Bregman divergence associated with a convex functional on $\mathcal{Y}$, then Eqn. 3.2 is the generalized Gauss-Newton matrix (Martens, 2014).

### 3.1 INDEPENDENCE METRIC

We now come to the heart of our paper: the construction of a metric inspired by the K-FAC approximation. Recall that K-FAC makes two approximations to obtain a tractable Fisher matrix: (1) it

assumes independence of activations and pre-activation derivatives in order to push the expectation inside the Kronecker product (Eqn. 2.7), and (2) it keeps only the diagonal blocks corresponding to individual layers. In this section, we develop a coordinate-free way to push the expectation inside the Kronecker product, thereby obtaining an approximate metric we term the *independence metric*. We later use this construction to develop approximate metrics for MLPs. In Section 3.2, we develop a coordinate-free version of the block-diagonal approximation. Combining both approximations yields the *K-FAC metric*, an intrinsic metric whose coordinate representation matches the K-FAC approximate Fisher matrix.

We begin by setting up the mathematical framework. To avoid tying ourselves to MLPs, we consider the more general setting of metrics on affine maps between affine spaces, but use notation which is suggestive of MLPs. We assume the following:

- Affine spaces $A$ and $Z$
- Affine space $W$ of affine transformations between $A$ and $Z$
- Metric $g$ on $Z$

Our first task is to formulate a coordinate-free analogue of the outer product of homogenized activations, $\bar{\mathbf{a}}_i \bar{\mathbf{a}}_i^\top$. Consider the evaluation map $\psi_a : W \to Z$ which is defined by evaluating $w \in W$ at $a \in A$. We compute the pushforward $\psi_{a*} : TW \to TZ$. Note that there is no need to specify particular points for the tangent spaces here since we are working with affine spaces (see Corollary C.2 in Appendix C). Let $\partial_w$ be a tangent vector on $W$ and $f$ be a smooth function on $Z$. Then,

$$\begin{aligned} \psi_{a*}(\partial_w)(f) &= \partial_w(f \circ \psi_a)(w) \\ &= (\partial_w f)(\psi_a(w)) \cdot \psi_a'(w) \\ &= (\partial_w f)(z) \cdot a \end{aligned}$$

This shows that the pushforward $\psi_{a*}$ is exactly multiplication by the element $a$. Hence, we can identify any $a \in A$ with its linear map $TW \to TZ$. Thus, this enables us to define the tensor product of two elements in $A$ as a mapping $a_1 \otimes a_2 : TW \times TW \to TZ \otimes TZ$:

$$(a_1 \otimes a_2)(\partial_{w_1}, \partial_{w_2}) = a_1(\partial_{w_1}) \otimes a_2(\partial_{w_2}).$$

We now introduce the central object of our study, inspired by the independence assumption for activations and pre-activation derivatives which led to Eqn. 2.7. For $w \in W$, define $g_{\mathrm{ind}}$ on $W$ to be

$$g_{\mathrm{ind}}(w) := \mathbb{E}[a \otimes a] \otimes \mathbb{E}[g(z)], \tag{3.3}$$

where the first expectation is over $A$ and the second one is over $Z$. Note that $\mathbb{E}[g(z)]$ is well defined because the affine structure of $Z$ allows us to identify the cotangent spaces at all points $z$. The following theorem, whose proof is given in Appendix E.2, shows that $g_{\mathrm{ind}}$ is indeed a metric on $W$.

**Theorem 3.4.** *Let $g$ be a metric on $Z$ and $\psi_a : W \to Z$ be the evaluation map. Then, $g_{\mathrm{ind}}$ as defined in Eqn. 3.3 is a metric on $W$. Moreover, if the expected pullback metric $\mathbb{E}_a[\psi_a^* g]$ is nondegenerate on $W$, then $g_{\mathrm{ind}}$ is also nondegenerate. From now on, we refer to $g_{\mathrm{ind}}$ as the* independence metric.

By direct computation, we obtain that the coordinate representation of $g_{\mathrm{ind}}$ matches the K-FAC approximation of the layerwise Fisher blocks:

**Proposition 3.5.** *Suppose that we choose coordinate systems for the affine spaces $A$, $Z$ and in these coordinates,*

$$[\![a]\!] = \mathbf{a}, \ [\![z]\!] = \mathbf{z}, \ [\![g(z)]\!] = \mathbf{G}(\mathbf{z}).$$

*Then the independence metric $g_{\mathrm{ind}}$ can be expressed as*

$$[\![g_{\mathrm{ind}}(w)]\!] = \mathbb{E}[\bar{\mathbf{a}} \bar{\mathbf{a}}^\top] \otimes \mathbb{E}[\mathbf{G}(\mathbf{z})].$$

*Remark* 3.6. In the context of MLPs (which we explain in greater detail subsequently) where $A = \mathcal{A}_{i-1}$, $Z = \mathcal{Z}_i$ and $W = \mathcal{W}_i$, the matrix $\mathbf{G}(z) = \mathbb{E}_\mathbf{y}[\mathcal{D}\mathbf{z}_i \mathcal{D}\mathbf{z}_i^\top]$ where the expectation is taken over output space. $\mathbb{E}[\mathbf{G}(z)]$ means we furthermore take the expectation over $\mathcal{Z}_i$. When we write $\mathbb{E}[\mathcal{D}\mathbf{z}_i \mathcal{D}\mathbf{z}_i^\top]$ in Eqn. 2.7, we implicitly take this to mean $\mathbb{E}[\mathbf{G}(z)]$.

### 3.2 K-FAC Metric

In this subsection, we formulate the layerwise independence approximation in a coordinate-free way, allowing us to define the K-FAC metric, whose coordinate representation matches the K-FAC approximation to the Fisher matrix. To do this, we use the independence metric developed in Section 3.1 to define the K-FAC metric for MLPs. Lastly, by viewing K-FAC as a metric on $\mathcal{W}$, we show how invariances of the K-FAC algorithm can be obtained in a natural and straightforward manner.

Consider a MLP with $L$ layers as described earlier in Section 3. For every $i \in \{1, \ldots, L\}$, we define the following maps

- $\psi^i_\xi : \mathcal{W}_i \to \mathcal{Z}_i$ which sends layerwise parameters $\omega_i$ to pre-activations $\zeta_i$ by evaluation at activations $\alpha_{i-1}$.
- $\varphi^i_\xi : \mathcal{Z}_i \to \mathcal{Y}$ which sends $\zeta_i$ to network outputs $\alpha_L = f(\xi, \omega)$.

The subscript $\xi$ is used to highlight the fact that all of these maps have an implicit dependence on network inputs $\xi$. Note that $\psi^i_\xi$ is a smooth map by definition. Now, observe that $\varphi^i_\xi$ is exactly the composition of network maps

$$\rho_L \circ \omega_L \circ \cdots \circ \omega_{i+1} \circ \rho_i : \mathcal{Z}_i \to \mathcal{Y}.$$

Since all activation functions $\rho_i$ are assumed to be smooth maps, it follows immediately that $\varphi^i_\xi$ is also a smooth map. Moreover, we define the map $\Psi^i_\xi : \mathcal{W}_i \to \mathcal{Y}$ to be composition $\Psi^i_\xi = \varphi^i_\xi \circ \psi^i_\xi$.

Let $g$ be a metric on $\mathcal{Y}$. Then, the pullback $(\varphi^i_\xi)^* g$ defines a metric on $\mathcal{Z}_i$. Now, if we take $A, Z, W$ in Section 3.1 to be

$$A = \mathcal{A}_{i-1}, \ Z = \mathcal{Z}_i, \ W = \mathcal{W}_i,$$

and the metric on $Z = \mathcal{Z}_i$ to be $(\varphi^i_\xi)^* g$, the independence metric on $\mathcal{W}_i$ here is

$$g^i_{\text{ind}}(\omega_i) = \mathbb{E}[\alpha_{i-1} \otimes \alpha_{i-1}] \otimes \mathbb{E}[(\varphi^i_\xi)^* g(\zeta_i)]. \tag{3.4}$$

Now, given metrics $g^i_{\text{ind}}$ on each $\mathcal{W}_i$, there is a natural "additive metric" defined on the Cartesian product $\mathcal{W} = \mathcal{W}_1 \times \cdots \times \mathcal{W}_L$. We defer the formalities and the sum "+" operation on metrics to Appendix B.5.

**Definition 3.7.** The K-FAC metric on the weight space $\mathcal{W}$ of a MLP is defined as

$$g_{\text{KFAC}}(\omega) = g^1_{\text{ind}}(\omega_1) + \cdots + g^L_{\text{ind}}(\omega_L),$$

where the sum is as defined in Eqn. B.2 in Appendix B.5 and each $g^i_{\text{ind}}$ is as given in Eqn. 3.4.

**Theorem 3.8** (See Appendix E.3 for proof)**.** *Let $g$ be a metric on $\mathcal{Y}$. Then, $g_{\text{KFAC}}$ given in Definition 3.7 is indeed a metric on the weight space $\mathcal{W}$ of an abstract MLP. Moreover, if we assume that the expected pullback of $g$,*

$$\mathbb{E}_\xi[(\Psi^i_\xi)^* g],$$

*under the map $\Psi^i_\xi : \mathcal{W}_i \to \mathcal{Y}$ is a nondegenerate metric on the layerwise weight space $\mathcal{W}_i$ for every $i$, then $g_{\text{KFAC}}$ is also nondegenerate.*

**Coordinate-Free Proof of Theorem 2.1.** We are now ready to provide a natural and straightforward proof of Theorem 2.1. In Appendix D, we show that networks $\mathcal{N}$ and $\mathcal{N}^\dagger$ correspond to two different choices of parameterizations for the same underlying abstract MLP. Hence, they must compute the same function.

For the latter assertion, assume that the metric $g$ on the output space $\mathcal{Y}$ in Theorem 3.8 is the output Fisher metric $g_{F,\text{out}}$ in Example 3.3. The pullback of this under $\varphi^i_\xi$ is given by

$$(\varphi^i_\xi)^* g_{F,\text{out}}(\zeta_i) = \mathbb{E}[d\mathcal{L}_{\zeta_i} \otimes d\mathcal{L}_{\zeta_i}].$$

Let us choose coordinate systems on $\mathcal{A}_{i-1}$ and $\mathcal{Z}_i$ with

$$[\![\alpha_{i-1}]\!] = \mathbf{a}_{i-1}, \ [\![\zeta_i]\!] = \mathbf{z}_i, \ [\![(\varphi^i_\xi)^* g_{F,\text{out}}(\zeta_i)]\!] = [\![\mathbb{E}[d\mathcal{L}_{\zeta_i} \otimes d\mathcal{L}_{\zeta_i}]]\!] = \mathbb{E}[\mathcal{D}\mathbf{z}_i \mathcal{D}\mathbf{z}_i^\top].$$

Then, by Proposition 3.5,

$$[\![g^i_{\text{ind}}(\omega_i)]\!] = \mathbb{E}[\bar{\mathbf{a}}_{i-1} \bar{\mathbf{a}}_{i-1}^\top] \otimes \mathbb{E}[\mathcal{D}\mathbf{z}_i \mathcal{D}\mathbf{z}_i^\top],$$

which is exactly $\hat{\mathbf{F}}(\mathbf{w})_{i,i}$ given earlier in Eqn. 2.7. Furthermore,

$$[\![g_{\text{KFAC}}(\omega)]\!] = [\![g_{\text{ind}}^1(\omega_1) + \cdots + g_{\text{ind}}^L(\omega_L)]\!]$$

is the matrix with diagonal blocks $\hat{\mathbf{F}}(\mathbf{w})_{i,i}$ and zeros everywhere else (see Appendix B.5). This is precisely $\hat{\mathbf{F}}(\mathbf{w})$ in Eqn. 2.8. Now, observe that

$$[\![g_{\text{KFAC}}(\omega)^{-1} dh(\omega)]\!] = [\![g_{\text{KFAC}}(\omega)^{-1}]\!] [\![dh(\omega)]\!] = \hat{\mathbf{F}}(\mathbf{w})^{-1} \nabla h(\mathbf{w}),$$

and hence the K-FAC update rule is simply a natural gradient update rule with respect to the K-FAC metric $g_{\text{KFAC}}$ for abstract MLPs. Suppose that $g_{\text{KFAC}}$ is a nondegenerate metric; which is true for example if the assumptions in the second assertion of Theorem 3.8 hold. Applying Theorem 3.2 shows that this update rule is invariant to any affine reparameterizations of the model. $\square$

## 4 APPLICATIONS TO CONVOLUTIONAL AND RECURRENT NETWORKS

In this section, we extend the preceding analysis to both convolutional and recurrent networks. Both cases are straightforward applications of our results in Section 3, highlighting the flexbility of our analysis. Due to space constraints, we give only sketches of the necessary constructions and defer the reader to Appendices F and G for full details.

### 4.1 CONVOLUTIONAL NETWORKS

We focus on a single convolution layer following the exposition given in Grosse & Martens (2016). Let $J$ be the number of input maps, $I$ be the number of output maps, $\Delta$ be the set of spatial offsets and $\mathcal{T}$ be the number of spatial locations. We write the convolution layer as a matrix multiplication

$$\begin{aligned} \mathbf{Z}_l &= \mathbf{W}_l \mathbf{A}_{l-1}^{\text{exp}} + \mathbf{b}_l \\ \mathbf{A}_l &= \phi_l(\mathbf{Z}_l), \end{aligned} \tag{4.1}$$

where $\mathbf{A}_{l-1}^{\text{exp}}$ is the $J|\Delta| \times |\mathcal{T}|$ matrix of expanded activations, $\mathbf{W}_l$ is the $I \times J|\Delta|$ weight matrix, $\mathbf{Z}_l$ is the $I \times |\mathcal{T}|$ matrix of pre-activations, $\phi_l$ is a nonlinear activation function and $\mathbf{b}_l$ is the bias vector. Adopting homogeneous coordinates for the various matrices, we can rewrite the above as

$$\begin{aligned} [\mathbf{Z}_l]_H &= [\mathbf{W}_l]_H [\mathbf{A}_{l-1}^{\text{exp}}]_H \\ [\mathbf{A}_l]_H &= \phi_l([\mathbf{Z}_l]_H). \end{aligned} \tag{4.2}$$

We briefly introduce the concept of a transformed convolution layer. For a convolution layer as defined in the above equation, the parameters $[\mathbf{W}_l]_H$ and the transformed parameters $[\mathbf{W}_l^\dagger]_H$ are related in the following way

$$[\mathbf{W}_l]_H = \mathbf{\Gamma}_l [\mathbf{W}_l^\dagger]_H (\mathbf{I} \otimes \mathbf{\Upsilon}_{l-1}), \tag{4.3}$$

where $\mathbf{\Gamma}_l$ and $\mathbf{\Upsilon}_{l-1}$ are invertible matrices. The activation functions $\phi_l$ and $\phi_l^\dagger$ are related through a standard affine change-of-basis as given in Eqn. 2.9.

For a convolutional network with $L$ convolution layers, the Fisher matrix is given by

$$\mathbf{F}(\mathbf{w}) = \mathbb{E}_{\mathbf{x},\mathbf{y}}[(\mathcal{D}\mathbf{w})(\mathcal{D}\mathbf{w})^\top],$$

where $\mathbf{w}$ is the vector concatenating all convolution layer weights. The K-FAC approximation is defined by setting all off-diagonal blocks to be zero and approximating the diagonal blocks as

$$\hat{\mathbf{F}}(\mathbf{w})_{l,l} = |\mathcal{T}|(\mathbb{E}_{\mathcal{T}}[\bar{\mathbf{a}}_{l-1}^{(:,t)}(\bar{\mathbf{a}}_{l-1}^{(:,t)})^\top] \otimes \mathbb{E}_{\mathcal{T}}[\mathcal{D}\mathbf{z}_l^{(t)}(\mathcal{D}\mathbf{z}_l^{(t)})^\top]). \tag{4.4}$$

Here, $\mathbf{a}_{l-1}^{(:,t)}$ stands for the expanded activations at each spatial location $t \in \mathcal{T}$ (the $J|\Delta|$-dimensional column vectors of the matrix $\mathbf{A}_{l-1}^{\text{exp}}$ in Eqn. 4.1) and $\mathbf{z}_l^{(t)}$ stands for the local pre-activations at $t \in \mathcal{T}$ (the $I$-dimensional column vectors of the matrix $\mathbf{Z}_l$ in Eqn. 4.1).

**Theorem 4.1.** *Let $\mathcal{N}$ be a convolutional network with parameter vector $\mathbf{w}$ and activation functions $\{\phi_l\}_{l=1}^L$. Suppose that we have activation functions $\{\phi_l^\dagger\}_{l=1}^L$ which are related to $\{\phi_l\}_{l=1}^L$ by standard change-of-basis transformations. Then, there exists a parameter vector $\mathbf{w}^\dagger$ such that the transformed network $\mathcal{N}^\dagger$ with parameter vector $\mathbf{w}^\dagger$ and activation functions $\{\phi_l\}_{l=1}^L$ computes the same function as $\mathcal{N}$. Furthermore, suppose that the Fisher matrix for the convolutional network is positive-definite, the K-FAC updates are equivalent, in the sense that the resulting networks compute the same function.*

**Proof Sketch (Full proof in Appendix F.3).** A coordinate-dependent proof of this result was given previously in Grosse & Martens (2016). Instead, we prove this intrinsically using the framework provided in Section 3. Let $\mathcal{A}_{l-1}^{\mathrm{exp}}$ be the affine space of expanded pre-activations at $t \in \mathcal{T}$, $\mathcal{Z}_l$ be the affine space of local pre-activations at $t \in \mathcal{T}$, and $\mathcal{W}_l$ be the layerwise weight space comprising of affine transformations between $\mathcal{A}_{l-1}^{\mathrm{exp}}$ and $\mathcal{Z}_l$. Now, let us take $A$, $Z$, $W$ in Section 3.1 to be $\mathcal{A}_{l-1}^{\mathrm{exp}}$, $\mathcal{Z}_l$, $\mathcal{W}_l$ respectively and the metric on $Z = \mathcal{Z}_l$ to be the pullback of the output Fisher metric. After summing over every spatial location $t \in \mathcal{T}$, the independence metric $g_{\mathrm{ind}}^l$ on $\mathcal{W}_l$ in coordinates, by Proposition 3.5, is exactly the expression given in Eqn. 4.4. Taking the sum of $g_{\mathrm{ind}}^l$'s then gives exactly the K-FAC approximation for convolutional networks. Finally, we know from Theorem 3.4 that each $g_{\mathrm{ind}}^l$ indeed defines a metric and so by Theorem 3.2, we can conclude that the K-FAC updates are exactly invariant to all affine reparameterizations of the model.

## 4.2 RECURRENT NETWORKS

As in the case of convolutional networks, we need not write out the full structure of a recurrent network. Rather, we focus solely on the recurrent computation since the object of our interest, the Fisher matrix, only involves recurrent weights. Let $T$ be the number of different time steps and $\mathbb{T} = \{1, \ldots, T\}$. We use $t$ to index the time step. Additionally, we assume that all sequences are of fixed length $T$. We consider the recurrent computation step

$$\mathbf{z}_t = \mathbf{W}\mathbf{a}_{t-1} + \mathbf{b}, \tag{4.5}$$

where $\mathbf{a}_{t-1}$ is an activation vector, $\mathbf{z}_t$ is a pre-activation vector, and $\mathbf{W}$ is a recurrent weight matrix. The transformed recurrent computation step is defined as

$$\mathbf{z}_t^\dagger = \mathbf{W}^\dagger \mathbf{a}_{t-1}^\dagger + \mathbf{b}^\dagger. \tag{4.6}$$

The relationship between transformed parameters $(\mathbf{W}^\dagger \ \mathbf{b}^\dagger)$ and original parameters $(\mathbf{W} \ \mathbf{b})$ is given by the standard change-of-basis formula as in Eqn. D.2.

The Fisher matrix for recurrent networks is defined as

$$\mathbf{F}(\bar{\mathbf{W}}) = \mathbb{E}_{\mathbf{x},\mathbf{y}}[\mathrm{vec}(\mathcal{D}\bar{\mathbf{W}}) \, \mathrm{vec}(\mathcal{D}\bar{\mathbf{W}})^\top],$$

and the K-FAC approximation is defined as

$$\hat{\mathbf{F}}(\bar{\mathbf{W}}) = T(\mathbb{E}_{\mathbb{T}}[\bar{\mathbf{a}}_{t-1}\bar{\mathbf{a}}_{t-1}^\top] \otimes \mathbb{E}_{\mathbb{T}}[\mathcal{D}\mathbf{z}_t \mathcal{D}\mathbf{z}_t^\top]). \tag{4.7}$$

**Theorem 4.2.** *Let $\mathcal{N}$ be a recurrent network with recurrent parameters $(\mathbf{W} \ \mathbf{b})$. Suppose that we have a recurrent network $\mathcal{N}^\dagger$ with recurrent parameters $(\mathbf{W}^\dagger \ \mathbf{b}^\dagger)$ and the relationship between $(\mathbf{W} \ \mathbf{b})$ and $(\mathbf{W}^\dagger \ \mathbf{b}^\dagger)$ is a standard change-of-basis transformation. Then, the networks $\mathcal{N}$ and $\mathcal{N}^\dagger$ compute the same function. Furthermore, if the Fisher matrix for recurrent networks is assumed to be positive-definite, then the K-FAC updates are equivalent, in the sense that the resulting networks compute the same function.*

**Proof Sketch (Full proof in Appendix G.3).** Let $\mathcal{A}$ be the affine space of local activations at each time step $t$, $\mathcal{Z}$ be the affine space of local pre-activations at each time step $t$, and $\mathcal{W}$ be the parameter space consisting of all affine transformations between $\mathcal{A}$ and $\mathcal{Z}$. Now, let us take $A$, $Z$, $W$ in Section 3.1 to be $\mathcal{A}$, $\mathcal{Z}$, $\mathcal{W}$ respectively and the metric on $Z = \mathcal{Z}$ to be the pullback of the output Fisher mertric. After summing over all time steps $t \in \mathbb{T}$, the independence metric $g_{\mathrm{ind}}$ on $\mathcal{W}$ in coordinates, by Proposition 3.5, is exactly the K-FAC approximation given in Eqn. 4.7. Finally, we know from Theorem 3.4 that $g_{\mathrm{ind}}$ indeed defines a metric and so by Theorem 3.2, we conclude that the K-FAC updates are exactly invariant to all affine reparameterizations of the model.

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

## A  TABLE OF NOTATIONS

|  | Coordinate-independent | Coordinate-dependent |
|---|---|---|
| inputs | $\xi$ | $\mathbf{x}$ |
| targets | $\upsilon$ | $\mathbf{y}$ |
| activations | $\alpha_i$ | $\mathbf{a_i}$ |
| homogenized activations |  | $\bar{\mathbf{a}}_i$ |
| pre-activations | $\zeta_i$ | $\mathbf{z_i}$ |
| layerwise parameters (and biases) | $\omega_i$ | $(\mathbf{W}_i\,\mathbf{b}_i)$ |
| homogenized layerwise parameters |  | $\bar{\mathbf{W}}_i$ |
| activation functions | $\rho_i$ | $\phi_i$ |
| network parameter | $\omega$ | $\mathbf{w}$ |
| network output | $f(\xi,\omega)$ | $f(\mathbf{x},\mathbf{w})$ |
| loss function | $\mathcal{L}(\upsilon,f(\xi,\omega))$ | $\mathcal{L}(\mathbf{y},f(\mathbf{x},\mathbf{w}))$ |
| objective function | $h(\omega)$ | $h(\mathbf{w})$ |
| predictive distribution | $P_{\upsilon|\xi}(\omega)$ | $P_{\mathbf{y}|\mathbf{x}}(\mathbf{w})$ |
| predictive distribution density function | $p(\upsilon|\xi,\omega)$ | $p(\mathbf{y}|\mathbf{x},\mathbf{w})$ |
| Fisher metric/matrix | $g_F(\omega)$ | $\mathbf{F}(\mathbf{w})$ |
| layerwise log-likelihood differential/gradient | $d\mathcal{L}_{\omega_i}$ | $\mathcal{D}\bar{\mathbf{W}}_i$ |
| log-likelihood differential/gradient | $d\mathcal{L}_{\omega}$ | $\mathcal{D}\mathbf{w}$ |
| K-FAC metric/matrix | $g_{\mathrm{KFAC}}(\omega)$ | $\hat{\mathbf{F}}(\mathbf{w})$ |
| tensor product/Kronecker product | $\otimes$ | $\otimes$ |
| input space | $\mathcal{X}$ |  |
| output space | $\mathcal{Y}$ |  |
| space of activations | $\mathcal{A}_i$ |  |
| space of pre-activations | $\mathcal{Z}_i$ |  |
| layerwise weight space | $\mathcal{W}_i$ |  |
| weight space of network | $\mathcal{W}$ |  |

Table 1: Since this paper involves the interplay of many coordinate-independent and coordinate-dependent mathematical objects, we summarize these notations here. Note that these notations are primarily for the setting of MLPs, the focus of the main body of the paper. The notations for convolutional networks and recurrent networks in Appendix F and G will be self-contained. As a general rule of thumb, we use boldface to symbolize coordinate-dependent objects and standard math font for coordinate-independent ones.

## B  MATHEMATICAL BACKGROUND

### B.1  VECTOR SPACES AND TENSOR ALGEBRA

Let $V$ be a vector space. The dual space $V^*$ of $V$ is the set of all linear functionals on $V$ and this space itself admits the structure of a vector space. The direct sum $U \oplus V$ of two vector spaces $U$ and $V$ is a vector space where the set structure is the Cartesian product $U \times V$ and the addition and multiplication is given by

$$(u_1,v_1) + (u_2,v_2) = (u_1 + v_1, u_2 + v_2)$$
$$c(u_1,v_1) = (cu_1,cv_1),$$

for $u_1,u_2 \in U$, $v_1,v_2 \in V$ and $c \in \mathbb{R}$. We now introduce tensors on vector spaces. A $k$-tensor $T$ on $V$ is a multilinear function

$$T : \underbrace{V \times \cdots \times V}_{k\text{ copies}} \to \mathbb{R},$$

We may think of $T$ as an element of the vector space $(V^*)^{\otimes k}$, the tensor product of the vector space $V^*$ with itself $k$-times. We delegate the definition of a tensor product of vector spaces to Appendix C.2. A $k$-tensor $T$ is symmetric if $T$ is a symmetric multilinear function. We work primarily with symmetric tensors in this paper.

### B.2    Canonical Isomorphisms

We describe the distinction between an isomorphism and a canonical isomorphism of vector spaces. An isomorphism between two vector spaces $U$ and $V$ is a bijection between $U$ and $V$ which preserves addition and scalar multiplication. A canonical isomorphism is a stronger concept, it is an isomorphism of vector spaces which is natural, in the sense that it does not depend on any choice of bases to define the isomorphism. For example, any two vector spaces of the same dimension are isomorphic to one another but the isomorphism may not be canonical. Consider a finite dimensional vector space $V$. There is an isomorphism between $V$ and its dual space $V^*$: given a choice of basis $\mathbf{e}_i$ for $V$, there is a dual basis $\mathbf{e}_i^*$ for $V^*$ and the map $\mathbf{e}_i \to \mathbf{e}_i^*$ is an isomorphism. However, this is not canonical as it depends on the choice of basis $\mathbf{e}_i$ for $V$. On the other hand, consider the evaluation map $\mathrm{ev}_v : V^* \to \mathbb{R}$ defined by $\varphi \mapsto \varphi(v)$ where $\varphi \in V^*$, $v \in V$. The mapping $v \mapsto \mathrm{ev}_v$ then defines a canonical isomorphism from $V$ to its double dual space $V^{**}$.

### B.3    Affine Algebra

A set $A$ is an affine space associated to the vector space $V$ if there is a mapping $A \times A \to V$ denoted by $(p, q) \in A \times A \mapsto \vec{pq} \in V$ satisfying the axioms

1. for any $p, q, r \in A$, $\vec{pr} = \vec{pq} + \vec{qr}$

2. for any $p \in A$ and for any $x \in V$ there is an unique $q \in A$ such that $x = \vec{pq}$.

Intuitively, an affine space may be thought of as a vector space with no privileged origin. Suppose that we choose an origin point $o \in A$ and let $\{\mathbf{e}_1, \ldots, \mathbf{e}_n\}$ be a basis for the associated vector space $V$. For any point $p \in A$, we can write $\vec{op} = \sum_{i=1}^{n} \mathbf{x}_i(p)\mathbf{e}_i$. Here, $\{\mathbf{x}_1, \ldots, \mathbf{x}_n\}$ is a set of coordinate functions, or more simply, a basis for $A$. If we have two bases $\{\mathbf{x}_1, \ldots, \mathbf{x}_n\}$ and $\{\mathbf{y}_1, \ldots, \mathbf{y}_n\}$, then they are related by $\mathbf{y} = \mathbf{Bx} + \mathbf{c}$ where $\mathbf{B} = [\mathbf{b}_{ij}]$ is an invertible $n \times n$ matrix and $\mathbf{c} = [\mathbf{c}_i]$ is a vector.

We now describe how to extend a change-of-basis on the affine space $A$ to the product space $A^K = A \times \cdots \times A$. Let $\iota$ and $\kappa$ be two choices of affine bases on $A$, then

$$\llbracket (a_1, \ldots, a_k) \rrbracket_\iota = (\bar{\mathbf{a}}_1, \ldots, \bar{\mathbf{a}}_k), \; \llbracket (a_1, \ldots, a_k) \rrbracket_\kappa = (\bar{\mathbf{a}}_1^\dagger, \ldots, \bar{\mathbf{a}}_k^\dagger),$$

where homogeneous coordinates are used for $\mathbf{a}_i$ and $\mathbf{a}_i^\dagger$. Now, suppose that the change-of-basis from $\iota$ to $\kappa$ is given by $(\mathbf{B} \; \mathbf{c})$ and denote

$$[\mathbf{B}]_H = \left[ \begin{array}{cc} \mathbf{B} & \mathbf{c} \\ \mathbf{0} & 1 \end{array} \right].$$

Then, we have

$$
\begin{aligned}
\left[ \begin{array}{c} \bar{\mathbf{a}}_1^\dagger \\ \vdots \\ \bar{\mathbf{a}}_k^\dagger \end{array} \right] &= \left[ \begin{array}{ccc} [\mathbf{B}]_H & & \mathbf{0} \\ & \ddots & \\ \mathbf{0} & & [\mathbf{B}]_H \end{array} \right] \left[ \begin{array}{c} \bar{\mathbf{a}}_1 \\ \vdots \\ \bar{\mathbf{a}}_k \end{array} \right] \\
&= (\mathbf{I} \otimes [\mathbf{B}]_H) \left[ \begin{array}{c} \bar{\mathbf{a}}_1 \\ \vdots \\ \bar{\mathbf{a}}_k \end{array} \right].
\end{aligned}
\tag{B.1}
$$

Thus, the induced change-of-basis on the product space $A^K$ is given by the matrix $\mathbf{I} \otimes [\mathbf{B}]_H$.

### B.4    Differentials, Pushforwards and Pullbacks

Let $\mathcal{M}$ be a smooth real manifold. For $p \in \mathcal{M}$, we denote the tangent space by $T_p\mathcal{M}$ and the corresponding dual space, the cotangent space by $T_p^*\mathcal{M}$. Given a smooth function $f : \mathcal{M} \to \mathbb{R}$, the differential $df(p) \in T_p^*\mathcal{M}$ is defined by

$$df(p)(X_p) = X_p(f), \; X_p \in T_p\mathcal{M}.$$

Let $(\mathbf{x}_1, \ldots, \mathbf{x}_n)$ be a coordinate system around $p \in \mathcal{M}$, the differential $df(p)$ can be expressed as

$$[\![df(p)]\!] = \begin{bmatrix} \frac{\partial f}{\partial \mathbf{x}_1} \\ \vdots \\ \frac{\partial f}{\partial \mathbf{x}_n} \end{bmatrix}.$$

We observe that this coordinate representation corresponds to the gradient $\nabla f$ (even though differentials and gradients are distinct objects for abstract manifolds).

For a smooth map $\varphi : \mathcal{M}_1 \to \mathcal{M}_2$ of manifolds, the pushforward $\varphi_* : T_p\mathcal{M}_1 \to T_{\varphi(p)}\mathcal{M}_2$ is defined by

$$\varphi_*(v)h = v(h \circ \varphi),$$

where $v \in T_p\mathcal{M}_1$ and $h : \mathcal{M}_2 \to \mathbb{R}$ is a smooth function on $\mathcal{M}_2$. If we suppose that $\mathcal{M}_1 = \mathbb{R}^n$ and $\mathcal{M}_2 = \mathbb{R}^m$ with $(\mathbf{x}_1, \ldots, \mathbf{x}_n)$ a coordinate system around $p$ and $(\mathbf{y}_1, \ldots, \mathbf{y}_m)$ a coordinate system around $\varphi(p)$, the pushforward $\varphi_* v$ can be represented as

$$[\![\varphi_* v]\!] = \begin{bmatrix} \frac{\partial \mathbf{y}_1}{\partial \mathbf{x}_1} & \cdots & \frac{\partial \mathbf{y}_1}{\partial \mathbf{x}_n} \\ \vdots & \ddots & \vdots \\ \frac{\partial \mathbf{y}_m}{\partial \mathbf{x}_1} & \cdots & \frac{\partial \mathbf{y}_m}{\partial \mathbf{x}_n} \end{bmatrix} \mathbf{v},$$

where $\mathbf{v} = [\![v]\!]$. This is exactly the Jacobian matrix $\mathbf{J}_\varphi$ of $\varphi$ and hence $[\![\varphi_* v]\!] = \mathbf{J}_\varphi \mathbf{v}$. This Jacobian-vector product corresponds to the directional derivative, and can be computed using forward mode automatic differentiation (Schraudolph, 2002).

The dual notion of the pushforward, the pullback $\varphi^* : T^*_{\varphi(p)}\mathcal{M}_2 \to T^*_p\mathcal{M}_1$ is defined in the following way

$$(\varphi^* u)(v) = u(\varphi_* v), \ u \in T^*_{\varphi(p)}\mathcal{M}_2.$$

With respect to the same coordinate systems chosen above, we can write $[\![\varphi^* u]\!] = \mathbf{J}_\varphi^\top \mathbf{u}$, where $\mathbf{u} = [\![u]\!]$. Numerically, we can compute $\mathbf{J}_\varphi^\top \mathbf{u}$ efficiently using reverse mode auto-differentiation (i.e., backpropagation).

## B.5 METRICS AND THEIR PROPERTIES

We introduce tensors on manifolds. A symmetric $k$-tensor $\sigma$ at the point $p \in \mathcal{M}$ is defined as a symmetric $k$-tensor on the tangent space $T_p\mathcal{M}$. Recall that this is a symmetric multilinear map on the $k$-fold product of $T_p\mathcal{M}$:

$$\sigma(p) : \underbrace{T_p\mathcal{M} \times \cdots \times T_p\mathcal{M}}_{k \text{ copies}} \to \mathbb{R}.$$

A metric on $\mathcal{M}$ is defined as a smoothly varying symmetric 2-tensor $g$ which is positive-semidefinite at every point $p \in \mathcal{M}$. If $g$ is nondegenerate, then this is just a usual Riemannian metric. However, in this paper, we use the term "nondegenerate" rather than "Riemannian" to describe such metrics.

Earlier in the paper, we pulled back metrics from the output space to the weight space of the network. Here, we define how this works for general tensors. Let $\varphi : \mathcal{M}_1 \to \mathcal{M}_2$ be a smooth map of manifolds and $\sigma$ be a symmetric $k$-tensor on $\mathcal{M}_2$ at $\varphi(p)$. The pullback $\varphi^* \sigma$ of $\sigma$ under $\varphi$ is a symmetric $k$-tensor on $\mathcal{M}_1$ defined as

$$\varphi^* \sigma(p)(v_1, \ldots, v_k) = \sigma(\varphi(p))(\varphi_* v_1, \ldots, \varphi_* v_k),$$

where $v_1, \ldots, v_k \in T_p\mathcal{M}_1$. In the case of metrics,

$$\varphi^* g_{\mathcal{M}_2}(p)(v_1, v_2) = g_{\mathcal{M}_2}(\varphi(p))(\varphi_* v_1, \varphi_* v_2).$$

If we suppose that the metric $g_{\mathcal{M}_2}$ is given by $\mathbf{G}_{\mathcal{M}_2}$ for a chosen coordinate system around $\varphi(p)$, then the pullback metric $\varphi^* g_{\mathcal{M}_2}$ on $\mathcal{M}_1$ around $p$ is given by

$$[\![\varphi^* g_{\mathcal{M}_2}(p)]\!] = \mathbf{J}_\varphi^\top \mathbf{G}_{\mathcal{M}_2} \mathbf{J}_\varphi,$$

where $\mathbf{J}_\varphi$ is the Jacobian of $\varphi$. While a metric always pulls back to a metric under a smooth map, the pullback of a nondegenerate metric can be degenerate as the pushforward map may have a non-trivial nullspace.

Lastly, we describe how to define metrics on product manifolds. Given metrics $g_{\mathcal{M}_1}$ and $g_{\mathcal{M}_2}$ on $\mathcal{M}_1$ and $\mathcal{M}_2$ respectively, we describe how to naturally define a metric on the product manifold $\mathcal{M}_1 \times \mathcal{M}_2$. For any point $(p, q) \in \mathcal{M}_1 \times \mathcal{M}_2$, there is a canonical isomorphism of tangent spaces:

$$T_{(p,q)}(\mathcal{M}_1 \times \mathcal{M}_2) \cong T_p\mathcal{M}_1 \oplus T_q\mathcal{M}_2.$$

The proof of this fact can be found in standard differential geometry literature (e.g. Lee (2003)) and so we do not elaborate further here. Hence, any vector $v \in T_{(p,q)}(\mathcal{M}_1 \times \mathcal{M}_2)$ can be written as a pair $(v_1, v_2)$ where $v_1 \in T_p\mathcal{M}_1$ and $v_2 \in T_q\mathcal{M}_2$. Now, we define the additive metric $g_{\mathcal{M}_1} + g_{\mathcal{M}_2}$ on $\mathcal{M}_1 \times \mathcal{M}_2$ as follows:

$$(g_{\mathcal{M}_1} + g_{\mathcal{M}_2})(p,q)(u,v) = g_{\mathcal{M}_1}(p)(u_1, v_1) + g_{\mathcal{M}_2}(q)(u_2, v_2). \tag{B.2}$$

If we choose a coordinate system around $(p, q)$ with the metrics $g_{\mathcal{M}_1}$, $g_{\mathcal{M}_2}$ represented by matrices $\mathbf{G}_{\mathcal{M}_1}$, $\mathbf{G}_{\mathcal{M}_2}$ respectively, then we have

$$[\![(g_{\mathcal{M}_1} + g_{\mathcal{M}_2})(p,q)]\!] = \begin{bmatrix} \mathbf{G}_{\mathcal{M}_1} & \mathbf{0} \\ \mathbf{0} & \mathbf{G}_{\mathcal{M}_2} \end{bmatrix},$$

which is a matrix with block diagonals $\mathbf{G}_{\mathcal{M}_1}, \mathbf{G}_{\mathcal{M}_2}$ and zero everywhere else. This construction generalizes easily to sums of more than two terms.

## C  ADDITIONAL MATHEMATICAL TECHNICALITIES

### C.1  TANGENT SPACE OF VECTOR SPACES AND AFFINE SPACES

**Theorem C.1.** *Let $V$ be a finite-dimensional vector space. For each point $p \in V$, there is a canonical isomorphism $V \to T_pV$. From now on, we suppress $p$ in $T_pV$ and write $TV$ when denoting tangent spaces of $V$.*

*Proof.* For any element $v \in V$, we can associate a tangent vector $D_v|_p$ at $p$ defined by

$$D_v|_p f = D_v f(p) = \frac{d}{dt}\Big|_{t=0} f(p + tv),$$

where $f$ is a smooth function on $V$. This gives the desired canonical isomorphism since the above construction involves no choice of basis. $\square$

**Corollary C.2.** *Let $A$ be an affine space and $V$ be its associated vector space. For each point $a \in A$, there is a canonical isomorphism between $T_aA$ and $V$. From now on, we suppress $a$ in $T_aA$ and write $TA$ when denoting tangent spaces of $A$.*

*Proof.* Note that specifying a point $a \in A$ naturally identifies $A$ with $V$. Then, applying the above theorem gives the desired result. $\square$

### C.2  TENSOR PRODUCT OF VECTOR SPACES

Let $U$ and $V$ be finite-dimensional vector spaces over the real numbers $\mathbb{R}$. Let $\mathcal{R}$ be the subspace of the free vector space $\mathbb{R}\langle U \times V \rangle$ (set of all finite formal linear combinations of elements of $U \times V$ with real coefficients) spanned by all elements of the following forms:

$$c(u, v) - (cu, v),$$
$$c(u, v) - (u, cv),$$
$$(u, v) + (u', v) - (u + u', v),$$
$$(u, v) + (u, v') - (u, v + v'),$$

for $u, u' \in U$, $v, v' \in V$, and $c \in \mathbb{R}$. The tensor product $U \otimes V$, is the quotient space $\frac{\mathbb{R}\langle U \times V \rangle}{\mathcal{R}}$ and the equivalence class of an element $(u, v)$ in $U \otimes V$ is denoted by $u \otimes v$.

We describe how the vector space of linear transformations between $U$ and $V$, denoted by $\mathrm{Hom}(U, V)$, may be thought of as tensor products. There is a canonical isomorphism

$$U^* \otimes V \to \mathrm{Hom}(U, V) \tag{C.1}$$

given by $\varphi \otimes v \mapsto \varphi(u)v$ where $\varphi \in U^*$. Another isomorphism of interest to us is

$$U^* \otimes V^* \to (U \otimes V)^*, \tag{C.2}$$

which is again canonical. To derive this isomorphism, given $\varphi \in U^*$, $\phi \in V^*$, consider the bilinear map $U \times V \to \mathbb{R}$ defined by

$$(u, v) \mapsto \varphi(u) \cdot \phi(v).$$

This induces an element on the tensor product $(U \otimes V)^*$. As such, we obtain an unique linear injection

$$U^* \otimes V^* \to (U \otimes V)^*.$$

Since all the vector spaces are finite-dimensional, we can conclude that this is an isomorphism.

We now use these facts to explain several ingredients in the proof of Theorem 3.2 in greater detail. The element $a$ is a linear map $TW \to TZ$ and by the above isomorphism, this means $a \in T^*W \otimes TZ$ from Eqn. C.1. By Eqn. C.2, the dual space of $(T^*W \otimes TZ)^* \cong TW \otimes T^*Z$. Using Eqn. C.1 again, elements of this space are maps $\lambda : T^*W \to T^*Z$.

## D  TWO PARAMETERIZATIONS OF COORDINATE-FREE MLP

We now show how the MLP with computation defined in Eqn. 2.6 and the transformed version in Eqn. 2.9 can be viewed as two different choices of parameterizations for the same underlying abstract MLP.

First, a choice of parameterization, or a coordinate system, for the abstract MLP is a choice of affine bases for all of the activation spaces $\mathcal{A}_1, \ldots, \mathcal{A}_{L-1}$, the pre-activation spaces $\mathcal{Z}_1, \ldots, \mathcal{Z}_L$, the input space $\mathcal{X}$ and the output space $\mathcal{Y}$ in the network. Observe that a choice of bases for $\mathcal{A}_{i-1}$ and $\mathcal{Z}_i$ naturally induces a basis for each $\mathcal{W}_i$, and therefore also for the full weight space $\mathcal{W}$. Let $\iota, \kappa$ be two different choices of parameterizations for the network. With respect to $\iota$, we write

$$[\![\alpha_{i-1}]\!]_\iota = \mathbf{a}_{i-1}, \ [\![\zeta_i]\!]_\iota = \mathbf{z}_i, \ [\![\omega_i]\!]_\iota = (\mathbf{W}_i \ \mathbf{b}_i), \ [\![\rho_i]\!]_\iota = \phi_i, \ [\![\alpha_i]\!]_\iota = \mathbf{a}_i,$$

and with respect to $\kappa$, we write

$$[\![\alpha_{i-1}]\!]_\kappa = \mathbf{a}_{i-1}^\ddagger, \ [\![\zeta_i]\!]_\kappa = \mathbf{z}_i^\ddagger, \ [\![\omega_i]\!]_\kappa = (\mathbf{W}_i^\ddagger \ \mathbf{b}_i^\ddagger), \ [\![\rho_i]\!]_\kappa = \phi_i^\ddagger, \ [\![\alpha_i]\!]_\kappa = \mathbf{a}_i^\ddagger.$$

Hence, we can rewrite Eqn. 3.1 in the parameterizations $\iota, \kappa$ as

$$
\begin{aligned}
\mathbf{z}_i &= \mathbf{W}_i \mathbf{a}_{i-1} + \mathbf{b}_i & \mathbf{z}_i^\ddagger &= \mathbf{W}_i^\ddagger \mathbf{a}_{i-1}^\ddagger + \mathbf{b}_i^\ddagger \\
\mathbf{a}_i &= \phi_i(\mathbf{z}_i), & \mathbf{a}_i^\ddagger &= \phi_i^\ddagger(\mathbf{z}_i^\ddagger).
\end{aligned}
\tag{D.1}
$$

The parameters $(\mathbf{W}_i \ \mathbf{b}_i)$ and $(\mathbf{W}_i^\ddagger \ \mathbf{b}_i^\ddagger)$ are related as follows

$$
\begin{aligned}
\mathbf{W}_i &= \mathbf{\Phi}_i \mathbf{W}_i^\ddagger \mathbf{\Omega}_{i-1} \\
\mathbf{b}_i &= \mathbf{\Phi}_i \mathbf{W}_i^\ddagger \boldsymbol{\gamma}_{i-1} + \mathbf{\Phi}_i \mathbf{b}_i^\ddagger + \boldsymbol{\tau}_i,
\end{aligned}
\tag{D.2}
$$

where $(\mathbf{\Omega}_{i-1} \ \boldsymbol{\gamma}_{i-1})$ is the change-of-basis from $\iota$ to $\kappa$ on $\mathcal{A}_{i-1}$ with $\mathbf{\Omega}_{i-1}$ an invertible matrix and $\boldsymbol{\gamma}_{i-1}$ a vector. Moreover, $(\mathbf{\Phi}_i \ \boldsymbol{\tau}_i)$ is the change-of-basis from $\kappa$ to $\iota$ on $\mathcal{Z}_i$ with $\mathbf{\Phi}_i$ an invertible matrix and $\boldsymbol{\tau}_i$ a vector. The activation functions $\phi_i$ and $\phi_i^\ddagger$ in Eqn. D.1 are related in the following way:

$$\mathbf{a}_i^\ddagger = \phi_i^\ddagger(\mathbf{z}_i^\ddagger) = \mathbf{\Omega}_i \phi_i(\mathbf{\Phi}_i \mathbf{z}_i^\ddagger + \boldsymbol{\tau}_i) + \boldsymbol{\gamma}_i, \tag{D.3}$$

where $(\mathbf{\Omega}_i \ \boldsymbol{\gamma}_i)$ is the change-of-basis from $\iota$ to $\kappa$ on $\mathcal{A}_i$.

To verify Eqn. D.2 and Eqn. D.3, we consider the the following commutative diagram (the top horizontal arrow is equal to the composition of maps given by the other three arrows) which relates the two parameterizations $\iota$ and $\kappa$ on the activation affine space $\mathcal{A}_{i-1}$ and the pre-activation affine space $\mathcal{Z}_i$:

$$
\begin{array}{ccc}
[\![\mathcal{A}_{i-1}]\!]_\iota & \xrightarrow{(\mathbf{W}_i \ \mathbf{b}_i)} & [\![\mathcal{Z}_i]\!]_\iota \\
{\scriptstyle (\mathbf{\Omega}_{i-1}, \boldsymbol{\gamma}_{i-1})} \downarrow & & \uparrow {\scriptstyle (\mathbf{\Phi}_i, \boldsymbol{\tau}_i)} \\
[\![\mathcal{A}_{i-1}]\!]_\kappa & \xrightarrow{(\mathbf{W}_i^\ddagger \ \mathbf{b}_i^\ddagger)} & [\![\mathcal{Z}_i]\!]_\kappa
\end{array}
\tag{D.4}
$$

Let $\mathbf{a}_{i-1} \in [\![\mathcal{A}_{i-1}]\!]_\iota$, this maps to $\mathbf{z}_i = \mathbf{W}_i \mathbf{a}_{i-1} + \mathbf{b}_i \in [\![\mathcal{Z}_i]\!]_\iota$ under the top horizontal arrow in Eqn. D.4. Now, mapping $\mathbf{a}_{i-1}$ under the composition of the other three arrows in Eqn. D.4, we obtain

$$
\begin{aligned}
\mathbf{a}_{i-1} &\mapsto \mathbf{\Omega}_i \mathbf{a}_{i-1} + \boldsymbol{\gamma}_{i-1} && \text{(apply left vertical arrow in Eqn. D.4)} \\
&\mapsto \mathbf{W}_i^\ddagger (\mathbf{\Omega}_i \mathbf{a}_{i-1} + \boldsymbol{\gamma}_{i-1}) + \mathbf{b}_i^\ddagger && \text{(apply bottom horizontal arrow in Eqn. D.4)} \\
&\mapsto \mathbf{\Phi}_i (\mathbf{W}_i^\ddagger \mathbf{\Omega}_i \mathbf{a}_{i-1} + \mathbf{W}_i^\ddagger \boldsymbol{\gamma}_{i-1}) + \mathbf{\Phi}_i \mathbf{b}_i^\ddagger + \boldsymbol{\tau}_i && \text{(apply right vertical arrow in Eqn. D.4)} \\
&= \mathbf{\Phi}_i \mathbf{W}_i^\ddagger \mathbf{\Omega}_{i-1} \mathbf{a}_{i-1} + \mathbf{\Phi}_i \mathbf{W}_i^\ddagger \boldsymbol{\gamma}_{i-1} + \mathbf{\Phi}_i \mathbf{b}_i^\ddagger + \boldsymbol{\tau}_i.
\end{aligned}
$$

This establishes the equality given in Eqn. D.2. For Eqn. D.3, we use the commutative diagram (this time, the bottom horizontal arrow is equal to the composition of the other three arrows):

$$
\begin{array}{ccc}
[\![\mathcal{Z}_i]\!]_\iota & \xrightarrow{\phi_i} & [\![\mathcal{A}_i]\!]_\iota \\
{\scriptstyle (\mathbf{\Phi}_i, \boldsymbol{\tau}_i)} \big\uparrow & & \big\downarrow {\scriptstyle (\mathbf{\Omega}_i, \boldsymbol{\gamma}_i)} \\
[\![\mathcal{Z}_i]\!]_\kappa & \xrightarrow{\phi_i^\ddagger} & [\![\mathcal{A}_i]\!]_\kappa
\end{array} \tag{D.5}
$$

Let $\mathbf{z}_i^\ddagger \in [\![\mathcal{Z}_i]\!]_\kappa$, this maps to $\mathbf{a}_i^\ddagger \in [\![\mathcal{A}_i]\!]_\kappa$ under $\phi_i^\ddagger$. Now, mapping $\mathbf{z}_i^\ddagger$ under the other three arrows in Eqn. D.5, we obtain

$$
\begin{aligned}
\mathbf{z}_i^\ddagger &\mapsto \mathbf{\Phi}_i \mathbf{z}_i^\ddagger + \boldsymbol{\tau}_i && \text{(apply left vertical arrow in Eqn. D.5)} \\
&\mapsto \phi_i (\mathbf{\Phi}_i \mathbf{z}_i^\ddagger + \boldsymbol{\tau}_i). && \text{(apply upper horizontal arrow in Eqn. D.5)} \\
&\mapsto \mathbf{\Omega}_i \phi_i (\mathbf{\Phi}_i \mathbf{z}_i^\ddagger + \boldsymbol{\tau}_i) + \boldsymbol{\gamma}_i && \text{(apply right vertical arrow in Eqn. D.5).}
\end{aligned}
$$

This establishes Eqn. D.3.

Lastly, we note that the equations in Eqn. D.1 can be rewritten in homogeneous coordinates:

$$
\begin{aligned}
\mathbf{z}_i &= \bar{\mathbf{W}}_i \bar{\mathbf{a}}_{i-1} & \mathbf{z}_i^\ddagger &= \bar{\mathbf{W}}_i \bar{\mathbf{a}}_{i-1}^\ddagger \\
\mathbf{a}_i &= \phi_i (\mathbf{z}_i), & \mathbf{a}_i^\ddagger &= \phi_i^\ddagger (\mathbf{z}_i^\ddagger) = \mathbf{\Omega}_i \phi_i (\mathbf{\Phi}_i \mathbf{z}_i^\ddagger + \boldsymbol{\tau}_i) + \boldsymbol{\gamma}_i.
\end{aligned}
$$

The left hand set of equations above is identical to the original MLP computation given in Eqn. 2.6. The right hand set of equations is identical to the transformed computation given in Eqn. 2.9. This completes the proof.

# E   PROOFS

## E.1   PROOF OF THEOREM 3.2

First, note that $g(\omega_k)^{-1} dh(\omega_k)$ is an intrinsically defined tangent vector on $\mathcal{W}$. The weight space $\mathcal{W}$ of a MLP is an affine space, and hence by Corollary C.2 in Appendix C, the tangent space of $\mathcal{W}$ at every point is canonically isomorphic to the vector space naturally associated to $\mathcal{W}$. Thus, the exponential map $\mathrm{Exp}_{\omega_k}$ corresponds to the above update rule. Since this construction did not require choosing an affine basis for $\mathcal{W}$, the algorithm is invariant to affine reparameterizations.

## E.2   PROOF OF THEOREM 3.4

Before providing the proof of the theorem, we need the following lemma.

**Lemma E.1.** *Let $g$ be a metric on $Z$ and $\psi_a : W \to Z$ be the evaluation map. Then, the pullback metric $\psi_a^* g$ on $W$ can be expressed as:*

$$
\psi_a^* g(w) = \mathbb{E}[a \otimes \phi \otimes a \otimes \phi],
$$

*where $\phi$ is a random covector.*

*Proof.* Given $z \in Z$, the metric $g$ admits the rank-1 decomposition

$$g(z) = \mathbb{E}[\phi \otimes \phi],$$

where $\phi$ is a covector and the expectation is over $Z$. This is akin to the more familiar case where any symmetric positive-semidefinite matrix admits a rank-1 spectral decomposition. Computing the pullback of $g$ under the map $\psi_a$ now gives

$$\begin{aligned}
\psi_a^* g(w) &= \psi_a^* \mathbb{E}[\phi \otimes \phi] \\
&= \mathbb{E}[\psi_a^*(\phi \otimes \phi)] \\
&= \mathbb{E}[\psi_a^* \phi \otimes \psi_a^* \phi],
\end{aligned} \tag{E.1}$$

where $z = \psi_a(w)$. We analyze the pullback $\psi_a^* \phi$. Let $\partial_w$ be a tangent vector on $W$. Then,

$$\begin{aligned}
(\psi_a^* \phi)(\partial_w) &= \phi((\psi_{a*})\partial_w) \\
&= \phi(\partial_w \cdot a) \\
&= (a \otimes \phi)(\partial_w),
\end{aligned}$$

which shows that $\psi_a^* \phi = a \otimes \phi$. Plugging this back into Eqn. E.1, we obtain

$$\psi_a^* g(w) = \mathbb{E}[a \otimes \phi \otimes a \otimes \phi],$$

which concludes the proof. $\qquad\square$

For the first assertion of the theorem, we need to check that both components $\mathbb{E}[a \otimes a]$ and $\mathbb{E}[g(z)]$ define symmetric positive-semidefinite 2-tensors. Recall that $a$ can be realized as a linear map from $TW$ to $TZ$. Then, the dual element $\lambda$ is a map from $T^*W$ to $T^*Z$. We refer to Appendix C.2 for a formal explanation of this. To check the positive-semidefinite property,

$$\mathbb{E}[a \otimes a](\lambda, \lambda) = \mathbb{E}[(a \otimes a)(\lambda \otimes \lambda)] \geq 0,$$

where the latter inequality is due to the fact that $a \otimes a$ is positive-semidefinite. Moreover, $a \otimes a$ is also symmetric and this property is preserved under expectations which implies that $\mathbb{E}[a \otimes a]$ is both symmetric and positive-semidefinite. For the second term $\mathbb{E}[g(z)]$ in $g_{\text{ind}}(w)$, the fact that $g$ is a metric on $Z$ means that $g(z)$, by definition, is a symmetric positive-semidefinite 2-tensor on $T^*Z$.

To establish the second assertion of the theorem, we need to show that both $\mathbb{E}[a \otimes a]$ and $\mathbb{E}[g(z)]$ are positive-definite. Suppose to the contrary that this is not true for $\mathbb{E}[a \otimes a]$. Then there exists $\lambda \in TW \otimes T^*Z$ (this is same as saying $\lambda$ is a linear map from $T^*W$ to $T^*Z$; refer to Appendix C.2 for further explanations) such that

$$\begin{aligned}
0 &= \mathbb{E}[a \otimes a](\lambda, \lambda) \\
&= \mathbb{E}[(a \otimes a)(\lambda \otimes \lambda)],
\end{aligned}$$

and hence $(a \otimes a)(\lambda \otimes \lambda) = 0$. Now, consider the element $\lambda \otimes \mu \in (TW \otimes T^*Z) \otimes TZ$ where $\mu \in TZ$. From the above lemma, we have

$$\psi_a^* g(w) = \mathbb{E}[a \otimes \phi \otimes a \otimes \phi]$$

Then, evaluating $\mathbb{E}_a[\psi_a^* g(w)]$ at $\lambda \otimes \mu$:

$$\begin{aligned}
\mathbb{E}_a[\psi_a^* g(w)](\lambda \otimes \mu, \lambda \otimes \mu) &= \mathbb{E}[a \otimes \phi \otimes a \otimes \phi](\lambda \otimes \mu \otimes \lambda \otimes \mu) \\
&= \mathbb{E}[\underbrace{(a \otimes a)(\lambda \otimes \lambda)}_{=0} \cdot (\phi \otimes \phi)(\mu \otimes \mu)] \\
&= 0,
\end{aligned}$$

This shows that $\mathbb{E}_a[\psi_a^* g]$ is not positive-definite which yields a contradiction as $\mathbb{E}_a[\psi_a^* g]$ was assumed to be a nondegenerate metric. The exact same argument can be applied to show that $\mathbb{E}[g(z)]$ is positive-definite. This gives us the desired result.

### E.3 Proof of Theorem 3.8

From Theorem 3.4, we know that $g_{\text{ind}}^i$ is a metric on $\mathcal{W}_i$. Since $g_{\text{KFAC}}$ is defined as the additive metric where each of the summands are $g_{\text{ind}}^i$, we can conclude that $g_{\text{KFAC}}$ is a metric. For the second assertion of the theorem, recall that $\Psi_\xi^i = \varphi_\xi^i \circ \psi_\xi^i$. By the functoriality property of pullback operations, we have

$$(\Psi_\xi^i)^* = (\psi_\xi^i)^* \circ (\varphi_\xi^i)^*.$$

Since $\mathbb{E}_\xi[(\Psi_\xi^i)^* g]$ was assumed to be nondegenerate, this implies that

$$\mathbb{E}_\xi[(\psi_\xi^i)^*((\varphi_\xi^i)^* g)],$$

is also nondegenerate. Then, from the second assertion of Theorem 3.4, we obtain that $g_{\text{ind}}^i$ is a nondegenerate metric on $\mathcal{W}_i$. Consequently, $g_{\text{KFAC}}$ is nondegenerate which concludes the proof.

## F Coordinate-Free K-FAC for Convolutional Networks

### F.1 Convolutional Networks

We provide a complete account of the convolutional network case sketched out earlier in Section 4.1. We begin by describing the convolution layer of a convolutional network in mathematical terms following Grosse & Martens (2016). We only consider convolution layers as the pooling and response normalization layers of a convolutional network typically do not contain (many) trainable weights. We then introduce the notion of a transformed convolution layer analogous to what was done in the case of MLPs. Lastly, we use the abstract linear algebra machinery developed in Section 2 to give a coordinate-free description of convolution layers.

#### F.1.1 Convolution Layers

We focus on a single convolution layer. A convolution layer $l$ takes as input a layer of activations $\mathbf{a}_{j,t}$, where $j \in \{1, \dots, J\}$ indexes the input map and $t \in \mathcal{T}$ indexes the spatial location. $\mathcal{T}$ here denotes the set of spatial locations, which we typically take to be a 2D-grid. We assume that the convolution is performed with a stride of 1 and padding equal to the kernel radius $R$, so that the set of spatial locations is shared between the input and output feature maps. This layer is parameterized by a set of weights $\mathbf{w}_{i,j,\delta}$ and biases $\mathbf{b}_i$, where $i \in \{1, \dots, I\}$ indexes the output map and $\delta \in \Delta$ indexes the spatial offset. The numbers of spatial locations and spatial offsets are denoted by $|\mathcal{T}|$ and $|\Delta|$ respectively. The computation at the convolution layer is given by

$$\mathbf{z}_{i,t} = \sum_{\delta \in \Delta} \mathbf{w}_{i,j,\delta} \mathbf{a}_{j,t+\delta} + \mathbf{b}_i. \tag{F.1}$$

The pre-activations $\mathbf{z}_{i,t}$ are then passed through a nonlinear activation function $\phi_l$. Analogous to feed-forward networks, the weight derivatives are computed using backpropagation:

$$\mathcal{D}\mathbf{w}_{i,j,\delta} = \sum_{t \in \mathcal{T}} \mathbf{a}_{j,t+\delta} \mathcal{D}\mathbf{z}_{i,t}.$$

Following Grosse & Martens (2016), we represent the convolution layer computation in Eqn. F.1 using matrix notation. To do this, we write the activations $\mathbf{a}_{j,t}$ as a $J \times |\mathcal{T}|$ matrix $\mathbf{A}_{l-1}$, the pre-activations $\mathbf{z}_{i,t}$ as a $I \times |\mathcal{T}|$ matrix $\mathbf{Z}_l$, the weights $\mathbf{w}_{i,j,\delta}$ as a $I \times J|\Delta|$ matrix $\mathbf{W}_l$ and the bias vector as $\mathbf{b}_l$. For the activation matrix $\mathbf{A}_{l-1}$, if we extract the patches surrounding each spatial location $t \in \mathcal{T}$ and flatten these patches into vectors where the vectors become columns of a matrix, we obtain a $J|\Delta| \times |\mathcal{T}|$ matrix which we denote by $\mathbf{A}_{l-1}^{\text{exp}}$. From now on, we refer to this matrix as the expanded activations. Finally, we can use these matrix notations to rewrite the computation in Eqn. F.1 as

$$\begin{aligned} \mathbf{Z}_l &= \mathbf{W}_l \mathbf{A}_{l-1}^{\text{exp}} + \mathbf{b}_l \\ \mathbf{A}_l &= \phi_l(\mathbf{Z}_l). \end{aligned} \tag{F.2}$$

For convenience purposes later, we adopt homogeneous coordinates for various matrices:

$$[\mathbf{A}_{l-1}^{\text{exp}}]_H = \begin{bmatrix} \mathbf{A}_{l-1}^{\text{exp}} \\ \mathbf{1} \end{bmatrix}, [\mathbf{Z}_{l-1}]_H = \begin{bmatrix} \mathbf{Z}_l \\ \mathbf{1} \end{bmatrix}, [\mathbf{W}_l]_H = \begin{bmatrix} \mathbf{W}_l & \mathbf{b}_l \\ \mathbf{0} & 1 \end{bmatrix}, [\mathbf{A}_l]_H = \begin{bmatrix} \mathbf{A}_l \\ \mathbf{1} \end{bmatrix}.$$

Hence, Eqn. F.2 can be rewritten as

$$
\begin{aligned}
[\mathbf{Z}_l]_H &= [\mathbf{W}_l]_H [\mathbf{A}_{l-1}^{\mathrm{exp}}]_H \\
[\mathbf{A}_l]_H &= \phi_l([\mathbf{Z}_l]_H),
\end{aligned}
\tag{F.3}
$$

where the activation function $\phi_l$ here ignores the homogeneous coordinate. We introduce the concept of a transformed convolution layer. For a convolution layer as defined in the above equation, the parameters $[\mathbf{W}_l]_H$ and the transformed parameters $[\mathbf{W}_l^\dagger]_H$ are related in the following way

$$
[\mathbf{W}_l]_H = \mathbf{\Gamma}_l [\mathbf{W}_l^\dagger]_H (\mathbf{I} \otimes \mathbf{\Upsilon}_{l-1}),
\tag{F.4}
$$

where $\mathbf{\Gamma}_l$ and $\mathbf{\Upsilon}_{l-1}$ are invertible matrices. The activation functions $\phi_l$ and $\phi_l^\dagger$ are related through a standard affine change-of-basis as given in Eqn. 2.9.

### F.1.2 ABSTRACT CONVOLUTION LAYERS

Just as in the coordinate-dependent case earlier, we focus on a single layer. An abstract convolution layer $l$ is defined as follows:

- Local activations at each spatial location $t \in \mathcal{T}$ are taken to be elements $\alpha_{l-1}$ in an affine space $\mathcal{A}_{l-1}$.

- Activations are taken to be elements $\alpha_{l-1}^{(:)}$ in $\mathcal{A}_{l-1}^{|\mathcal{T}|}$, (i.e. the direct product of $\mathcal{A}$, $|\mathcal{T}|$ times). (The superscripts are meant to be suggestive of Python slicing notation.)

- Expanded activations at $t \in \mathcal{T}$ are taken to be elements $\alpha_{l-1}^{(:,t)}$ in $\mathcal{A}_{l-1}^{|\Delta|}$. The full expanded activations are taken to be elements $\alpha_{l-1}^{(:,:)}$ in $\mathcal{A}_{l-1}^{|\Delta| \otimes |\mathcal{T}|}$.

- Local pre-activations at $t \in \mathcal{T}$ are taken to be elements $\zeta_l^{(t)}$ in an affine space $\mathcal{Z}_l$.

- Pre-activations are taken to be elements $\zeta_l^{(:)}$ in $\mathcal{Z}_l^{|\mathcal{T}|}$.

- Layerwise parameters are affine transformations $\omega_l$ between $\mathcal{A}_{l-1}^{|\Delta|}$ and $\mathcal{Z}_l$. The collection of these transformations is an affine space in its own right which we denote by $\mathcal{W}_l$ and refer to as the layerwise weight space. If we apply $\omega_l$ pointwise, this can be extended to a map

$$
\mathcal{A}_{l-1}^{|\Delta| \otimes |\mathcal{T}|} \to \mathcal{Z}_l^{|\mathcal{T}|}.
$$

The computation for this abstract layer is

$$
\begin{aligned}
\zeta_l^{(t)} &= \omega_l(\alpha_{l-1}^{(:,t)}) \\
\alpha_l &= \rho_l(\zeta_l^{(t)}),
\end{aligned}
$$

where $\rho_l$ is a fixed nonlinear activation function and $\alpha_l$ are the $l$-th layer local activations defined in exactly the same manner as $\alpha_{l-1}$.

We choose affine bases on $\mathcal{A}_{l-1}$, $\mathcal{Z}_l$, and $\mathcal{A}_l$. A basis on $\mathcal{A}_{l-1}$ naturally induces a basis for $\mathcal{A}_{l-1}^{|\Delta|}$. Consequently, this gives a basis also for the layerwise parameter space $\mathcal{W}_l$. Let $\iota$, $\kappa$ be two such choices. With respect to $\iota$, we write

$$
[\![\alpha_{l-1}^{(:,t)}]\!]_\iota = \mathbf{a}_{l-1}^{(:,t)}, \ [\![\zeta_l^{(t)}]\!]_\iota = \mathbf{z}_l^{(t)}, \ [\![\omega_l]\!]_\iota = (\mathbf{W}_l \ \mathbf{b}_l), \ [\![\rho_l]\!]_\iota = \phi_l, \ [\![\alpha_l]\!]_\iota = \mathbf{a}_l,
$$

and with respect to $\kappa$, we write

$$
[\![\alpha_{l-1}^{(:,t)}]\!]_\kappa = (\mathbf{a}_{l-1}^{(:,t)})^\ddagger, \ [\![\zeta_l^{(t)}]\!]_\kappa = (\mathbf{z}_l^{(t)})^\ddagger, \ [\![\omega_l]\!]_\kappa = (\mathbf{W}_l^\ddagger \ \mathbf{b}_l^\ddagger), \ [\![\rho_l]\!]_\kappa = \phi_l^\ddagger, \ [\![\alpha_l]\!]_\kappa = \mathbf{a}_l^\ddagger.
$$

Note that $\mathbf{a}_{l-1}^{(:,t)}$ are $J|\Delta|$-dimensional column vectors of the expanded activations matrix $\mathbf{A}_{l-1}^{\mathrm{exp}}$, $\mathbf{z}_l^{(t)}$ are $I$-dimensional column vectors of pre-activations matrix $\mathbf{Z}_l$ and $\mathbf{a}_l$ are $J$-dimensional column vectors of activations matrix $\mathbf{A}_l$.

Now, suppose that $(\mathbf{\Omega}_{l-1} \ \gamma_{l-1})$ is the change-of-basis from $\iota$ to $\kappa$ on $\mathcal{A}_{l-1}$ and $(\mathbf{\Phi}_l \ \tau_l)$ is the change-of-basis from $\kappa$ to $\iota$ on $\mathcal{Z}_l$. If we denote

$$
[\mathbf{\Omega}_{l-1}]_H = \begin{bmatrix} \mathbf{\Omega}_{l-1} & \gamma_{l-1} \\ \mathbf{0} & 1 \end{bmatrix}, \ [\mathbf{\Phi}_l]_H = \begin{bmatrix} \mathbf{\Phi}_l & \tau_l \\ \mathbf{0} & 1 \end{bmatrix},
$$

then by the affine change-of-basis formula for direct products (Eqn. B.1), $\mathbf{I} \otimes [\boldsymbol{\Omega}_{l-1}]_H$ defines the change-of-basis from $\iota$ to $\kappa$ on $\mathcal{A}_{l-1}^{|\Delta|}$. The parameters $[\mathbf{W}_l]_H$ and $[\mathbf{W}_l^\ddagger]_H$ are related as follows:

$$[\mathbf{W}_l]_H = [\boldsymbol{\Phi}_l]_H [\mathbf{W}_l^\ddagger]_H (\mathbf{I} \otimes [\boldsymbol{\Omega}_{l-1}]_H)$$

By taking $\boldsymbol{\Upsilon}_{l-1}$ and $\boldsymbol{\Gamma}_l$ in Eqn. F.4 to be $\boldsymbol{\Upsilon}_{l-1} = [\boldsymbol{\Omega}_{l-1}]_H$ and $\boldsymbol{\Gamma}_l = [\boldsymbol{\Phi}_l]_H$, we can conclude that a convolution layer and its transformed version simply correspond to two different choices of parameterizations for the same underlying abstract convolution layer.

## F.2 KRONECKER FACTORS FOR CONVOLUTION

We review the Kronecker Factors for Convolution method due to Grosse & Martens (2016) which is a version of K-FAC for convolutional networks. The network architecture to consider is a convolutional network with $L$ convolution layers. First, let $\mathbf{w}$ be the concatenation of all trainable parameters $\bar{\mathbf{W}}_l$,

$$\mathbf{w} = [\text{vec}(\bar{\mathbf{W}}_1)^\top \ \text{vec}(\bar{\mathbf{W}}_2)^\top \ \ldots \ \text{vec}(\bar{\mathbf{W}}_L)^\top]^\top.$$

For an input-target pair $(\mathbf{x}, \mathbf{y})$, the Fisher matrix for this network is

$$\mathbf{F}(\mathbf{w}) = \mathbb{E}_{\mathbf{x},\mathbf{y}}[(\mathcal{D}\mathbf{w})(\mathcal{D}\mathbf{w})^\top],$$

where $\mathcal{D}\mathbf{w}$ is the log-likelihood gradient and the expectation is taken over the model's predictive distribution $P_{\mathbf{y}|\mathbf{x}}(\mathbf{w})$ for $\mathbf{y}$ and over the data distribution for $\mathbf{x}$. The diagonal blocks $\mathbf{F}(\mathbf{w})_{l,l}$ of $\mathbf{F}(\mathbf{w})$ are

$$\mathbf{F}(\mathbf{w})_{l,l} = \mathbb{E}[\text{vec}(\mathcal{D}\bar{\mathbf{W}}_l) \text{vec}(\mathcal{D}\bar{\mathbf{W}}_l)^\top].$$

We are ready now to present the K-FAC approximation for convolutional networks. For a particular layer $l$, we define the K-FAC approximation $\hat{\mathbf{F}}(\mathbf{w})_{l,l}$ to $\mathbf{F}(\mathbf{w})_{l,l}$ as:

$$\hat{\mathbf{F}}(\mathbf{w})_{l,l} = |\mathcal{T}|(\mathbb{E}_\mathcal{T}[\bar{\mathbf{a}}_{l-1}^{(:,t)}(\bar{\mathbf{a}}_{l-1}^{(:,t)})^\top] \otimes \mathbb{E}_\mathcal{T}[\mathcal{D}\mathbf{z}_l^{(t)}(\mathcal{D}\mathbf{z}_l^{(t)})^\top]), \tag{F.5}$$

where $\bar{\mathbf{a}}_{l-1}^{(:,t)}$ is the homogeneous notation for $\mathbf{a}_{l-1}^{(:,t)}$. The K-FAC approximation $\hat{\mathbf{F}}(\mathbf{w})$ to $\mathbf{F}(\mathbf{w})$ is the matrix with diagonal blocks $\hat{\mathbf{F}}(\mathbf{w})_{l,l}$ as given above and zeros everywhere else.

*Remark* F.1. Unlike MLPs where K-FAC is derived from assuming only the statistical independence of activations and pre-activation derivatives, convolution layers admit weight sharing and additional assumptions are necessary to derive the approximation $\hat{\mathbf{F}}(\mathbf{w})_{l,l}$ in Eqn. F.5. We refer to Grosse & Martens (2016) for extensive details on how these approximations are derived.

We devote the remainder of this section to deriving invariance properties of K-FAC for convolutional networks. We refer the reader to Grosse & Martens (2016) for other aspects of the K-FAC algorithm on convolutional networks, such as implementation details and experimental results.

## F.3 COORDINATE-FREE K-FAC FOR CONVOLUTIONAL NETWORKS

We begin by considering an abstract convolutional network with $L$ convolution layers. Let $\mathcal{X}$ and $\mathcal{Y}$ denote the input and output spaces of this network respectively. Recall that the layerwise weight space $\mathcal{W}_l$ is the space of affine transformations between $\mathcal{A}_{l-1}^{|\Delta|}$ and $\mathcal{Z}_l$. The weight space of this network is the direct product of all layerwise weight spaces

$$\mathcal{W} = \mathcal{W}_1 \times \cdots \times \mathcal{W}_L.$$

Given an input $\xi \in \mathcal{X}$ and parameter $\omega = (\omega_1, \ldots, \omega_L) \in \mathcal{W}$, denote the network output by $f(\xi, \omega)$. Now, for every $l \in \{1, \ldots, L\}$, define the following maps

- $\psi_\xi^l : \mathcal{W}_l \to \mathcal{Z}_l^{|\mathcal{T}|}$ which sends layerwise parameters $\omega_l$ to pre-activations $\zeta_l^{(:)}$ by evaluating local activations $\alpha_{l-1}$ across every spatial location $t \in \mathcal{T}$

- $\varphi_\xi^l : \mathcal{Z}_l^{|\mathcal{T}|} \to \mathcal{Y}$ which sends pre-activations $\zeta_l^{(:)}$ to $f(\xi, \omega)$

Again, $\psi_\xi^l$ is trivially a smooth map from its definition. The map $\varphi_\xi^l$ includes all operations in convolutional networks such as max-pooling and response normalization. We make an assumption here that all these operations are smooth. (While this is not the case for common operations such as ReLU and max-pooling, we conjecture that the non-smooth case can be addressed by taking limits of smooth functions.) Finally, we define the map $\Psi_\xi^l : \mathcal{W}_l \to \mathcal{Y}$ as the composition $\Psi_\xi^l = \varphi_\xi^l \circ \psi_\xi^l$.

Let $g$ be a metric on $\mathcal{Y}$ and consider the pullback $(\varphi_\xi^l)^* g$ restricted to a single spatial location $t$ which we denote by $(\varphi_{\xi,t}^l)^* g$. More concretely, this metric is computed by assuming components of the tangent vector at all other spatial locations are zero. Now, let us take $A, Z, W$ in Section 3.1 to be

$$A = \mathcal{A}_{l-1}^{|\Delta|}, \; Z = \mathcal{Z}_l, \; W = \mathcal{W}_l,$$

and the metric on $Z = \mathcal{Z}_l$ to be $(\varphi_{\xi,t}^l)^* g$. Summing over every spatial location $t \in \mathcal{T}$, the independence metric on $\mathcal{W}_l$ here is

$$g_{\mathrm{ind}}^l(\omega_l) = |\mathcal{T}|(\mathbb{E}_\mathcal{T}[\alpha_{l-1}^{(:,t)} \otimes \alpha_{l-1}^{(:,t)}] \otimes \mathbb{E}_\mathcal{T}[(\varphi_{\xi,t}^l)^* g(\zeta_l^{(t)})]). \tag{F.6}$$

**Definition F.2.** The K-FAC metric on the weight space $\mathcal{W}$ of an abstract convolutional network is defined as

$$g_{\mathrm{KFAC}}(\omega) = g_{\mathrm{ind}}^1(\omega_1) + \cdots + g_{\mathrm{ind}}^L(\omega_L),$$

where the sum above is as defined in Eqn. B.2 and each $g_{\mathrm{ind}}^l$ is as given in Eqn. F.6.

**Theorem F.3.** *Let $g$ be a metric on $\mathcal{Y}$. Then, $g_{\mathrm{KFAC}}$ given in Definition F.2 is indeed a metric on the weight space $\mathcal{W}$ of an abstract convolutional network. Moreover, if we assume that the expected pullback of $g$ restricted to a single spatial location $t \in \mathcal{T}$,*

$$\mathbb{E}_\xi[(\Psi_{\xi,t}^l)^* g],$$

*under the map $\Psi_{\xi,t}^l : \mathcal{W}_l \to \mathcal{Y}$ is a nondegenerate metric on the layerwise weight space $\mathcal{W}_l$ for every $l$, then $g_{\mathrm{KFAC}}$ is also nondegenerate.*

*Proof.* The proof of this theorem mirrors the proof given earlier for Theorem 3.8. By Theorem 3.4, we know that

$$\mathbb{E}[\alpha_{l-1}^{(:,t)} \otimes \alpha_{l-1}^{(:,t)}] \otimes \mathbb{E}[(\varphi_{\xi,t}^l)^* g(\zeta_l^{(t)})],$$

is a metric on $\mathcal{W}_l$. Since taking expectation over the set of spatial locations $\mathcal{T}$ and multiplying by the scale factor $|\mathcal{T}|$ preserves the metric properties, we obtain that $g_{\mathrm{ind}}^l$ in Eqn. F.6 defines a metric on $\mathcal{W}_l$. Consequently, $g_{\mathrm{KFAC}}$ determines a metric on $\mathcal{W}$. To prove the latter assertion, note that by the functorial property of pullback operations,

$$\mathbb{E}_\xi[(\psi_{\xi,t}^l)^* ((\varphi_{\xi,t}^l)^* g)]$$

is nondegenerate. Using the second assertion of Theorem 3.4 yields that $g_{\mathrm{ind}}^l$ is nondegenerate which implies that this is true also for $g_{\mathrm{KFAC}}$. $\qquad\square$

We conclude this section with the proof of Theorem 4.1. Our proof is coordinate-free and given in exactly the same manner as the proof of Theorem 2.1 at the end of Section 3.

**Coordinate-Free Proof of Theorem 4.1.** As shown earlier in Section F.1.2, each convolution layer of $\mathcal{N}$ and $\mathcal{N}^\dagger$ correspond to two different choices of parameterizations for the same underlying abstract convolution layer. Hence, $\mathcal{N}$ and $\mathcal{N}^\dagger$ must compute the same function.

Assume that the metric $g$ on the output space $\mathcal{Y}$ in Theorem F.3 is the output Fisher metric $g_{F,\mathrm{out}}$. For each spatial location $t \in \mathcal{T}$, the pullback under $\varphi_{\xi,t}^l$ is

$$(\varphi_{\xi,t}^l)^* g_{F,\mathrm{out}}(\zeta_l^{(t)}) = \mathbb{E}[d\mathcal{L}_{\zeta_l^{(t)}} \otimes d\mathcal{L}_{\zeta_l^{(t)}}].$$

Now, choose coordinate systems on $\mathcal{A}_{l-1}$ and $\mathcal{Z}_l$. This induces coordinates for $\mathcal{A}_{l-1}^{|\Delta|}$ and we write

$$[\![\alpha_{l-1}^{(:,t)}]\!] = \mathbf{a}_{l-1}^{(:,t)}, \; [\![\zeta_l^{(t)}]\!] = \mathbf{z}_l^{(t)}, \; [\![(\varphi_{\xi,t}^l)^* g_{F,\mathrm{out}}(\zeta_l^{(t)})]\!] = [\![\mathbb{E}[d\mathcal{L}_{\zeta_l^{(t)}} \otimes d\mathcal{L}_{\zeta_l^{(t)}}]]\!] = \mathbb{E}[\mathcal{D}\mathbf{z}_l^{(t)} (\mathcal{D}\mathbf{z}_l^{(t)})^\top].$$

Using Proposition 3.5, the independence metric in Eqn. F.6 can be expressed in coordinates as follows

$$[\![g_{\mathrm{ind}}^l(\omega_l)]\!] = |\mathcal{T}|(\mathbb{E}_{\mathcal{T}}[\bar{\mathbf{a}}_{l-1}^{(:,t)}(\bar{\mathbf{a}}_{l-1}^{(:,t)})^\top] \otimes \mathbb{E}_{\mathcal{T}}[\mathcal{D}\mathbf{z}_l^{(t)}(\mathcal{D}\mathbf{z}_l^{(t)})^\top]),$$

which is exactly $\hat{\mathbf{F}}(\mathbf{w})_{l,l}$ given earlier in Eqn. F.5. Furthermore, $[\![g_{\mathrm{KFAC}}(\omega)]\!]$ is exactly the K-FAC approximation $\hat{\mathbf{F}}(\mathbf{w})$. Thus, the K-FAC update rule is simply a natural gradient update rule with respect to the K-FAC metric $g_{\mathrm{KFAC}}$ for abstract convolutional networks. Lastly, if $g_{\mathrm{KFAC}}$ is a nondegenerate metric; which is true for example if the assumptions in the second assertion of Theorem F.3 hold, then we can conclude that these updates are invariant to any affine reparameterizations of the model. $\qquad\square$

# G   COORDINATE-FREE K-FAC FOR RECURRENT NETWORKS

In this section, we provide a complete account of the recurrent network case sketched out earlier in Section 4.2. We first give a mathematical formulation of the recurrent computation step of these networks in both coordinate-dependent and coordinate-independent scenarios. We proceed to give the Kronecker factorization of the Fisher matrix for recurrent networks and then state the invariance theorem for this optimization method. Lastly, we prove the invariance theorem in the same way we did for MLPs and convolutional networks previously.

## G.1   RECURRENT NETWORKS

As in the case of convolutional networks in Section F, it is not necessary to write out the full structure of a recurrent network. Rather, we focus on the recurrent computation since the central object of our interest, the Fisher matrix for recurrent networks, only involves recurrent weights.

### G.1.1   COMPUTATIONAL STEP

Let $T$ be the number of different time steps and $\mathbb{T} = \{1, \ldots, T\}$. We use $t$ to index the time step. Throughout, we assume that all sequences are of fixed length $T$. For an input $\mathbf{x}_t$ at every $t$, the recurrent network maps this to an output $\mathbf{o}_t$. Essentially, the network maps input sequences $\mathbf{x} = (\mathbf{x}_1, \ldots, \mathbf{x}_T)$ to output sequences $\mathbf{o} = (\mathbf{o}_1, \ldots, \mathbf{o}_T)$. The computation, at every $t$, is

$$\mathbf{z}_t = \mathbf{W}\mathbf{a}_{t-1} + \mathbf{b}$$
$$\mathbf{z}_t' = \mathbf{z}_t + \mathbf{V}\mathbf{x}_t$$
$$\mathbf{a}_t = \phi(\mathbf{z}_t')$$

where $\mathbf{a}_{t-1}$ is an activation vector, $\mathbf{z}_t$ is a pre-activation vector, $\mathbf{W}$ is a recurrent weight matrix, $\mathbf{V}$ is a weight matrix, $\mathbf{b}$ is a recurrent bias vector, and $\phi$ is a fixed nonlinear activation function. For the remainder of this section, we focus on the first equation

$$\mathbf{z}_t = \mathbf{W}\mathbf{a}_{t-1} + \mathbf{b}, \tag{G.1}$$

which represents the recurrent computation step. The latter two equations can be handled by the previous K-FAC analysis for MLPs given in Section 3. The transformed recurrent computation step is defined as

$$\mathbf{z}_t^\dagger = \mathbf{W}^\dagger \mathbf{a}_{t-1}^\dagger + \mathbf{b}^\dagger. \tag{G.2}$$

The relationship between transformed parameters $(\mathbf{W}^\dagger \ \mathbf{b}^\dagger)$ and original parameters $(\mathbf{W} \ \mathbf{b})$ is given by the standard change-of-basis formula as in Eqn. D.2.

### G.1.2   ABSTRACT RECURRENT NETWORK

We now describe an abstract recurrent network formally.

- Local activations at each time step $t$ are elements $\alpha_t$ in an affine space $\mathcal{A}$
- Activations are elements $\alpha = \{\alpha_t\}_{t \in \mathbb{T}}$ in the affine space $\mathcal{A}^T$
- Local pre-activations at each $t$ are elements $\zeta_t$ in an affine space $\mathcal{Z}$
- Pre-activations are elements $\zeta = \{\zeta_t\}_{t \in \mathbb{T}}$ in the affine space $\mathcal{Z}^T$

- Parameters are affine transformations $\omega$ between $\mathcal{A}$ and $\mathcal{Z}$. The collection of these transformations is an affine space in its own right which we denote by $\mathcal{W}$ and refer to as the weight space
- Network inputs and outputs at each $t$ are elements $\xi_t$, $\upsilon_t$ in affine spaces $\mathcal{X}$, $\mathcal{Y}$ respectively. The input and output spaces are $\mathcal{X}^T$ and $\mathcal{Y}^T$ respectively; furthermore, elements here are written as $\xi = \{\xi_t\}_{t\in\mathbb{T}}$ and $\upsilon = \{\upsilon_t\}_{t\in\mathbb{T}}$.

For every $t$, the abstract recurrent computation step is

$$\zeta_t = \omega(\alpha_{t-1}).$$

A choice of parameterization for the abstract recurrent network consists of choosing affine bases for $\mathcal{A}$, $\mathcal{Z}$, $\mathcal{X}$ and $\mathcal{Y}$. Since we have bases for $\mathcal{A}$ and $\mathcal{Z}$, this induces a natural basis for $\mathcal{W}$. If we use exactly the same change-of-basis analysis given in Appendix D, then the recurrent network with computation given by Eqn. G.1 and the transformed version in Eqn. G.2 correspond to two different parameterizations of the same abstract recurrent network.

## G.2 K-FAC FOR RECURRENT NETWORKS

We review the recent Kronecker factorization for recurrent networks method in Martens et al. (2018). Recall that for every time step $t$, the recurrent computation can be written as

$$\mathbf{z}_t = \bar{\mathbf{W}}\bar{\mathbf{a}}_{t-1},$$

where $\bar{\mathbf{W}} = [\mathbf{W} \ \mathbf{b}]$ and $\bar{\mathbf{a}}_{t-1}^\top = [\mathbf{a}_{t-1}^\top \ 1]^\top$. Using backpropagation, the log-likelihood gradient is given by $\mathcal{D}\mathbf{z}_t\bar{\mathbf{a}}_{t-1}^\top$. The total contribution to the gradient across all $t$ is the sum

$$\mathcal{D}\bar{\mathbf{W}} = \sum_{t=1}^{T} \mathcal{D}\mathbf{z}_t\bar{\mathbf{a}}_{t-1}^\top.$$

For an input-target pair $(\mathbf{x}, \mathbf{y})$, the Fisher matrix $\mathbf{F}(\bar{\mathbf{W}})$ for recurrent networks is defined as

$$\mathbf{F}(\bar{\mathbf{W}}) = \mathbb{E}_{\mathbf{x},\mathbf{y}}[\text{vec}(\mathcal{D}\bar{\mathbf{W}})\,\text{vec}(\mathcal{D}\bar{\mathbf{W}})^\top].$$

Finally, the K-FAC approximation $\hat{\mathbf{F}}(\bar{\mathbf{W}})$ to $\mathbf{F}(\bar{\mathbf{W}})$ for recurrent networks is defined as

$$\hat{\mathbf{F}}(\bar{\mathbf{W}}) = T(\mathbb{E}_{\mathbb{T}}[\bar{\mathbf{a}}_{t-1}\bar{\mathbf{a}}_{t-1}^\top] \otimes \mathbb{E}_{\mathbb{T}}[\mathcal{D}\mathbf{z}_t\mathcal{D}\mathbf{z}_t^\top]). \tag{G.3}$$

*Remark* G.1. As in the case of convolution layers, there is weight sharing in recurrent networks (across time here instead of spatial locations) and so it is not enough to just assume statistical independence between activations and pre-activation derivatives to make the K-FAC approximation here. We defer the reader to Martens et al. (2018) for detailed explanations on how the K-FAC approximation is derived for recurrent networks.

We now proceed to the last section of this paper to give a coordinate-free proof of Theorem 4.2. The method of proof mirrors exactly the proofs given previously for Theorem 2.1 and Theorem 4.1.

## G.3 COORDINATE-FREE K-FAC FOR RECURRENT NETWORKS

Given an input $\xi = \{\xi_t\}_{t\in\mathbb{T}} \in \mathcal{X}^T$ and parameter $\omega \in \mathcal{W}$, denote the network output by $f(\xi, \omega)$. For a specific time step $t$, consider the following maps:

- $\psi_{\xi,t} : \mathcal{W} \to \mathcal{Z}$ which sends parameters $\omega$ to pre-activations $\zeta_t$ by evaluation at activations $\alpha_{t-1}$
- $\varphi_{\xi,t} : \mathcal{Z} \to \mathcal{Y}$ which sends $\zeta_t$ to outputs

In addition, we define the map $\Psi_{\xi,t} : \mathcal{W} \to \mathcal{Y}$ as the composition $\Psi_{\xi,t} = \varphi_{\xi,t} \circ \psi_{\xi,t}$.

Let $g$ be a metric on $\mathcal{Y}$. The pullback $\varphi_{\xi,t}^* g$ then defines a metric on $\mathcal{Z}$. Now, we take $A$, $Z$, $W$ in Section 3.1 to be

$$A = \mathcal{A}, \ Z = \mathcal{Z}, \ W = \mathcal{W},$$

and the metric on $Z = \mathcal{Z}$ to be $\varphi_{\xi,t}^* g$. Summing over all time steps $t \in \mathbb{T}$, we make the following definition which arises from the independence metric in Section 3.2:

**Definition G.2.** The K-FAC metric on the weight space $\mathcal{W}$ of an abstract recurrent network is defined as

$$g_{\mathrm{KFAC}}(\omega) = T(\mathbb{E}_{\mathbb{T}}[\alpha_{t-1} \otimes \alpha_{t-1}] \otimes \mathbb{E}_{\mathbb{T}}[\varphi^*_{\xi,t} g(\zeta_t)]). \tag{G.4}$$

**Theorem G.3.** *Let $g$ be a metric on $\mathcal{Y}$. Then, $g_{\mathrm{KFAC}}$ given in Definition G.2 is a metric on the weight space $\mathcal{W}$ of an abstract recurrent network. Moreover, if we assume that the expected pullback of $g$,*

$$\mathbb{E}_\xi[\Psi^*_{\xi,t} g],$$

*under the smooth map $\Psi_{\xi,t} : \mathcal{W} \to \mathcal{Y}$ is a nondegenerate metric, then $g_{\mathrm{KFAC}}$ is also nondegenerate.*

*Proof.* The proof of this theorem is analogous to the proofs of Theorem 3.8 and Theorem F.3. From Theorem 3.4, we have that

$$\mathbb{E}[\alpha_{t-1} \otimes \alpha_{t-1}] \otimes \mathbb{E}[\varphi^*_{\xi,t} g(\zeta_t)]$$

is a metric on $\mathcal{W}$. Since this remains true after taking expectation over the set of time steps $\mathbb{T}$ and multiplying by the scale factor $T$, we can conclude that $g_{\mathrm{KFAC}}$ is a metric on $\mathcal{W}$. For the nondegeneracy statement, using the functorial property of pullbacks, we know that

$$\mathbb{E}_\xi[\psi^*_{\xi,t}(\varphi^*_{\xi,t} g)]$$

is nondegenerate. Then, $g_{\mathrm{KFAC}}$ is nondegenerate by the second assertion of Theorem 3.4. □

**Coordinate-Free Proof of Theorem 4.2.** We conclude this paper with a coordinate-free proof of Theorem 4.2. As mentioned at the end of Section G.1.2, $\mathcal{N}$ and $\mathcal{N}^\dagger$ correspond to two different choices of parameterizations for the same underlying abstract recurrent network and so they must compute the same function.

Assume that the metric $g$ on $\mathcal{Y}$ in Theorem G.3 above is the output Fisher metric $g_{F,\mathrm{out}}$. Then, the pullback under $\varphi_{\xi,t}$ is

$$\varphi^*_{\xi,t} g_{F,\mathrm{out}}(\zeta_t) = \mathbb{E}[d\mathcal{L}_{\zeta_t} \otimes d\mathcal{L}_{\zeta_t}].$$

Now, choose coordinate systems for $\mathcal{A}$ and $\mathcal{Z}$. We can write

$$[\![\alpha_{t-1}]\!] = \mathbf{a}_{t-1}, \ [\![\zeta_t]\!] = \mathbf{z}_t, \ [\![\varphi^*_{\xi,t} g_{F,\mathrm{out}}(\zeta_t)]\!] = \mathbb{E}[\mathcal{D}\mathbf{z}_t \mathcal{D}\mathbf{z}_t^\top].$$

By Proposition 3.5, the K-FAC metric in Eqn. G.4 can be represented in these chosen coordinates as

$$[\![g_{\mathrm{KFAC}}(\omega)]\!] = T(\mathbb{E}_{\mathbb{T}}[\bar{\mathbf{a}}_{t-1} \bar{\mathbf{a}}_{t-1}^\top] \otimes \mathbb{E}_{\mathbb{T}}[\mathcal{D}\mathbf{z}_t \mathcal{D}\mathbf{z}_t^\top]),$$

which is exactly the K-FAC approximation $\hat{\mathbf{F}}(\bar{\mathbf{W}})$ in Eqn. G.3. Thus, the K-FAC update rule is a natural gradient update with respect to the K-FAC metric $g_{\mathrm{KFAC}}$ for abstract recurrent networks. If we suppose that $g_{\mathrm{KFAC}}$ is nondegenerate, then these updates are invariant to any affine reparameterizations of the model. □

