# OpenReview forum: "A Coordinate-Free Construction of Scalable Natural Gradient"
_ICLR.cc/2020/Conference — Reject_

### Official Review · AnonReviewer3 · 2019-10-14
**Official Blind Review #3**

**Rating:** 3

**Review:**

This paper is concerned with tractable (approximate) forms of natural gradient updates for neural networks, in particular with the recent K-FAC approximation, which applies a set of approximation (layer-wise independence, Kronecker structure for affine maps) in order to obtain a Hessian that can be computed and inverted efficiently. K-FAC has been introduced for MLPs, and has previously been generalized to convolutional and certain recurrent NNs.

The stated goal of this paper is to provide a mathematical re-formulation of K-FAC in terms of Riemannian metrics. While K-FAC has been developed as approximation to the exact natural gradient update, they come up with a different Riemannian metric, definition of space, etc., such that in the end, K-FAC is the exact natural gradient for that. The authors here also obtain a more precise answer to invariance properties and, given some heavy maths, what they claim to be more elegant proofs of previously known properties of K-FAC.

The paper uses very heavy math, well "over my head" and likely most ICLR attendees. Along with me, they'll ask the obvious question of what this is good for. As far as I can see, there is nothing really new being proposed here in terms of practical consequences. The authors also do not make much effort to explain why their viewpoint is useful, say to obtain practically relevant insights in future work. So, as far as I am concerned, I do not see why this work should be of much relevance to ICLR, which is not an abstract maths conference.

A final comment is that people have for a very long time tried to use second-order optimization for MLPs. The aspect that always was tricky there, is that SGD is *stochastic*, and the second-order info is hard to estimate from a mini-batch. The sets of approximations of K-FAC are pretty extreme, but they may just be needed to make things work in the end, because they may stabilize that "stochastic inverse Fisher info matrix" enough to not make the optimization process fail altogether. Now, all theoretical arguments, like "invariance to this and that", always ignore the crucial fact that you are talking about a stochastic estimate over a mini-batch, and your theory is always for E_x[...] "being the truth". It is not, it is just over a small mini-batch. I am not saying the additional theoretical insight from this work here (over previous K-FAC work), whatever it may be in the end, is not useful. I am just saying I'd be a lot more confident if the authors would specifically address the stochastic property.

**Experience Assessment:**

I have published in this field for several years.

**Review Assessment: Checking Correctness Of Derivations And Theory:**

I did not assess the derivations or theory.

**Review Assessment: Checking Correctness Of Experiments:**

N/A

**Review Assessment: Thoroughness In Paper Reading:**

I read the paper at least twice and used my best judgement in assessing the paper.

---

> ### Author Response · Authors · 2019-11-14
> **Response to Reviewer #3**
>
> We would like to thank the reviewer for carefully reading our manuscript and their feedback. We address them below:
>
> First, please see our response to Reviewer 2 concerning the practical significance of our work. K-FAC is essentially the only scalable and low-overhead method available for applying second-order optimization to large neural networks (eg. ImageNet classifiers), so developing a systematic understanding of its intrinsic properties is important. Since neural networks and optimization are among core topics at ICLR and invariances are among the topics neural network optimization researchers are most interested in understanding; our work is certainly aligned with the scope of ICLR.
>
> “Heavy math”:
>
> We agree that the math is indeed heavy for a machine learning audience but differential geometry and abstract linear + affine algebra are the appropriate tools for the job. As we explained in Section 2 of the paper, the reason why invariance properties of natural gradient are obtained automatically is due to the fact that it admits an intrinsic differential-geometric construction. The invariances of K-FAC, as an approximation to natural gradient, should be derived in exactly the same way; as opposed to manual matrix-algebra manipulations which offer little insight to the true nature of the algorithm.
>
> Moreover, we wrote the paper as self-contained as possible; we provided background sections in the Appendix covering all the necessary mathematical concepts and tools needed for this paper.
>
> “Stochastic estimates of K-FAC/Fisher matrix”:
>
> Our current theory in this paper is based on taking the expectation over inputs on the output Fisher matrix (actually our setup covered pullback metrics which is much more general). We agree that it would be interesting to investigate the stochastic version; however, we believe this to be outside the scope of our paper. The recent paper [1] demonstrates that taking empirical estimates of the Fisher matrix is not only a very different mathematical object than the Fisher matrix itself; but it can also behave quite differently even on simple optimization problems over toy models.
>
> [1] Kunstner, Frederik, Lukas Balles, and Philipp Hennig. "Limitations of the Empirical Fisher Approximation." arXiv preprint arXiv:1905.12558 (2019).

---

> > ### Comment · AnonReviewer3 · 2019-11-14
> > **Response to author feedback**
> >
> > I stick with my assessment, which seems pretty consistent with the others, except that the first reviewer even pointed out several flaws (buried in the mass of maths, apparently).
> >
> > I maintain that this paper has very little chance of creating any downstream impact, because it just offers another way to look at K-FAC, it does not give improvements, or even lead the way to them. The maths for this way is so heavy I doubt it will lead to any real progress. AI/ML is not pure math, we need to focus on tangible improvements in practice, or explain why heuristics work.
> >
> > I'd encourage the authors to use their new viewpoint grounded in pure math to develop some useful extension or improvement of K-FAC and demonstrate that in serious experiments, and resubmit elsewhere.

---

### Official Review · AnonReviewer2 · 2019-10-25
**Official Blind Review #2**

**Rating:** 3

**Review:**

This paper analyzes the invariance properties of the K-FAC algorithm by reconstructing the algorithm in a coordinate-free way where the neural network is viewed as a series of affine mappings alternating with nonlinear activation functions. It converts the original metric into an approximate metric, whose coordinate representation matches the K-FAC approximation. So K-FAC can be viewed as the exact natural gradient under the new metric rather than an approximation under the Fisher metric.

Why is the invariance important? How does the proposed framework help us develop better algorithms? Without empirical studies, it is not easy to see the significance of this work.

**Experience Assessment:**

I do not know much about this area.

**Review Assessment: Checking Correctness Of Derivations And Theory:**

I did not assess the derivations or theory.

**Review Assessment: Checking Correctness Of Experiments:**

N/A

**Review Assessment: Thoroughness In Paper Reading:**

I made a quick assessment of this paper.

---

> ### Author Response · Authors · 2019-11-14
> **Response to Reviewer #2**
>
> We would like to thank the reviewer for their comments and feedback. We address them below:
>
> “Why is invariance important?”
>
> Applying transformations to a neural network; for example, replacing logistic activation functions with tanh, whitening the inputs or activations and batch normalization, has lead to significant speed-ups in optimization performance. The de-facto optimization method, Stochastic Gradient Descent (SGD), is not invariant to any of these transformations. It makes sense for us to use an optimization algorithm which is invariant to such transformations. In this paper, we provided a complete analysis of the invariance properties of K-FAC for a variety of network architectures; demonstrating that it is fully invariant (not just first-order) to all affine reparameterizations, which covers many of the common tricks and transformations used when training neural networks.
>
> “Significance + practical impact?”
>
> Our current work is meant to be purely a theoretical study of the K-FAC algorithm and its invariance properties. The ultimate objective of adopting the intrinsic approach (framing everything in coordinate-free mathematical objects) is that it allows us to easily reason about invariance properties of the algorithm. Our framework allows a unified and cohesive way of studying K-FAC for different architectures and metrics (not limited to just the Fisher matrix/metric) all at once. All of the invariance proofs can be encapsulated into basic theorems and so we would not be stuck with heavy matrix algebra manipulations every time we wanted to derive invariance properties. Indeed, prior to our work, proving K-FAC invariance properties for MLPs and ConvNets in [1] and [2] respectively involved pages of tedious calculations (some of which were completely redundant with each other).
>
> From the practical standpoint, K-FAC is one of the few scalable approximations to natural gradient, and this approximation has been applied in many contexts besides deep learning optimization, such as Bayesian neural nets, continual learning, and network pruning. Analyzing the invariances carefully, as we have done in this paper, can help inform all of these use cases.
>
> [1] Martens, James, and Roger Grosse. "Optimizing neural networks with kronecker-factored approximate curvature."
> [2] Grosse, Roger B., and James Martens. "A Kronecker-factored approximate Fisher matrix for convolution layers."

---

### Official Review · AnonReviewer1 · 2019-10-25
**Official Blind Review #1**

**Rating:** 3

**Review:**

Natural gradient (NG) has been proven efficient in statistical learning, and one of its attractive properties is being invariant under smooth transformations of the parameter space. Computing NG is often difficult as one has to derive the Fisher matrix and its inverse. K-FAC offers an approximate method for approximating the NG with the risk of losing the invariant property.

This present paper offers a different approach: rather than approximating the exact NG directly as K-FAC does, it views the approximate K-FAC natural gradient as the exact "natural gradient" under the K-FAC metric. This is an interesting idea and the paper goes on to show that the new "exact" NG under the K-FAC metric (rather than the Fisher-Rao metric) is invariant under certain affine transformations.

I find the paper extremely hard to follow. My main concern is that many concepts and math objects are at risk of being not mathematically rigorous or well defined. For example, equation (2.5) is not correct, this is not a genuine update based on exponential map. Similarly, the update equation in the first graph of page 4 doesn't make sense as \mathcal{W} is an abstract manifold and the subtract here is not defined. The way paper is written has a high risk of causing confusion between coordinate-dependent and coordinate-free objects. On page 3, line -5, I have trouble understanding the abstract distribution P_{v|\psi}(\omega). On page 7, I don't understand what "evaluating $w$ at $a$" means, and even $a$ here isn't defined yet I think. In equation (2.3), $-w_k$ should be $w_k$ (i.e. no minus sign). There are many other typos and confusions that I opt not to  point out here.

In summary, the main issue of the paper in its current form is that its presentation isn't clear with many typos. I suggest the authors to re-write the paper carefully so that it is more accessible.

I know that this paper focuses on theoretical properties of NG, but it would also be good if the authors offer some discussion on the use of their method in practice.




**Experience Assessment:**

I have read many papers in this area.

**Review Assessment: Checking Correctness Of Derivations And Theory:**

I assessed the sensibility of the derivations and theory.

**Review Assessment: Checking Correctness Of Experiments:**

N/A

**Review Assessment: Thoroughness In Paper Reading:**

I read the paper at least twice and used my best judgement in assessing the paper.

---

> ### Author Response · Authors · 2019-11-14
> **Response to Reviewer #1**
>
> We would like to thank the reviewer for carefully reading our paper and their comments. We address them below:
>
> Concerning the following mathematical mistakes:
>
> - Equation 2.5 and update equation at the top of page 4: We have corrected this in the revised version. However, we want to stress that the update rule in Theorem 3.2, the one central to our study, is correct: subtraction is well-defined on an affine space and on an affine space, all tangent spaces at different points are canonically isomorphic and so the update makes sense. This theorem is what enables us to show that our constructed K-FAC metric is exactly invariant (not just first-order) to all affine reparameterizations of the network.
>
> - The abstract distribution P_{\upsilon|\psi}(\omega); we think of (\psi,\upsilon) as an abstract input-target pair where \psi and \upsilon are on input and output spaces which are themselves smooth manifolds. We have added this explanation in the revised version.
>
> - Evaluating w at a on page 7: A, Z are affine space and W is the affine space whose elements comprise of affine maps between A and Z. The “a” here refers to an element of A so the evaluation map W → Z takes an element “w” in W; itself an affine mapping from A to Z, evaluates it at the element “a”, to give an element in Z. We have added this explanation also in the revised version.
>
> We would like to mention that we spent lots of time and effort on how to present this work to a machine learning conference venue within the page limit; for example, choosing good notations which machine learning researchers are familiar with (as opposed to ones more conventional in mathematics) as well as devoting entire sections in the Appendix to explain all the necessary mathematical tools and concepts to ensure that the paper is as self-contained as possible. As mentioned at the end of the introduction, we realize that the paper frequently goes back and forth between coordinate-independent and dependent objects. Hence, to aid the reader, we provided a complete table summarizing the interplay between the two scenarios in Appendix A.

---

### Decision · Program_Chairs · 2019-12-19

**Decision:**

Reject

**Comment:**

The authors analyze the natural gradient algorithm for training a neural net from a theoretical perspective and prove connections to the K-FAC algorithm. The paper is poorly written and contains no experimental evaluation or well established implications wrt practical significance of the results.